

# Improving JULES Soil Moisture Estimates through 4D-En-Var Hybrid Assimilation of COSMOS-UK Soil Moisture Observations

Ramesh Visweshwaran[1,2], Elizabeth Cooper[3], and Sarah L Dance[1,2,4]

[1]Department of Meteorology, University of Reading, Reading, UK.
[2]National Centre for Earth Observation (NCEO), Reading, UK.
[3]UK Centre for Ecology & Hydrology (UKCEH), Wallingford, UK.
[4]Department of Mathematics & Statistics, University of Reading, Reading, UK.

**Correspondence:** Ramesh Visweshwaran (v.ramesh@reading.ac.uk)

**Abstract.** Accurate soil moisture estimates are essential for effective management and operational planning in various applications, including flood and drought response. However, soil moisture values derived from land surface models often exhibit significant deviations from in-situ observations. Data assimilation combines model information with observations to enhance prediction accuracy. Previous studies have typically focused on either estimating the initial soil moisture state or optimizing

Pedotransfer Function (PTF) constants, which link soil texture to the hydraulic properties of the land surface models. In contrast, in this study, we employ a novel approach by performing joint state-parameter assimilation for the JULES model. We optimized both the PTF constants and the initial soil moisture conditions simultaneously. Using Four-Dimensional Ensemble Variational hybrid data assimilation, we ingested field-scale soil moisture observations from the Cosmic-ray Soil Moisture Monitoring Network across 16 diverse sites in the UK. The results demonstrate that joint state-parameter assimilation signif-

icantly enhances the accuracy of soil moisture estimates, improving the average Kling Gupta Efficiency values from 0.33 to 0.72 across different soil characteristics. These findings indicate that our proposed joint state-parameter assimilation framework holds great potential for enhanced predictive accuracy in land surface models.

## 1 Introduction

Soil moisture plays a significant role in land surface processes due to its strong influence on various components of the water

cycle. Soil moisture controls the partitioning of incoming rainfall into surface runoff, percolation, and subsequent subsurface runoff (Entekhabi et al., 2010). Thus, accurate monitoring of soil moisture is essential for proper water resource management, including irrigation, and responses to floods and droughts (Roy et al., 2021; Visweshwaran et al., 2022a; Hou and Pu, 2024; Sadhwani and Eldho, 2024; de Roos et al., 2024).

Soil moisture conditions can be monitored using in-situ instruments, remote sensing from both ground- and space-based

platforms, or estimated by forcing land surface models (LSMs) and hydrological models with meteorological observations. Remotely sensed observations from sensors such as Soil Moisture and Ocean Salinity (SMOS; Kerr et al. (2001)), Soil Moisture Active Passive (SMAP; Entekhabi et al. (2010)), Advanced Scatterometer (ASCAT; Wagner et al. (2013)), Sentinel 1 (Torres et al., 2012), and Advanced Microwave Scanning Radiometer-2 (AMSRE; Njoku et al. (2003)) determine soil moisture





conditions across a wide area but provide information only at the surface level due to low penetration capacity (Albergel et al.,

2012). Conversely, in-situ observations from networks like SMOSMANIA (Calvet et al., 2007) and SMOSREX (de Rosnay et al., 2006) provide accurate soil moisture conditions at both surface and subsurface levels. However, these observations are limited to point locations at selected sites, offering less spatial coverage compared to satellite-based methods. In recent years, new approaches have been developed to measure soil moisture at the field scale. The Cosmic-ray Soil Moisture Monitoring Network (COSMOS-UK), provided by the UK Centre for Ecology and Hydrology (UKCEH), uses cosmic ray neutron sensors

(CNRS) to monitor soil moisture conditions. These sensors capture data over a horizontal footprint of approximately 25 to 30 hectares and measure to a vertical depth of up to 80 cm with high accuracy (Cooper et al., 2021b; Stanley, 2023). These observations are available a 30-minute temporal scale, providing continuous, real-time monitoring of soil moisture variations.

Land surface models can estimate soil moisture conditions at various soil depths, including surface, subsurface, and deeper soil layers, and across a range of spatial and temporal scales, typically down to a few meters spatially and at sub-daily to

weekly intervals temporally. However, these estimates are uncertain. These uncertainties can stem from errors in the forcing variables, such as rainfall and temperature data,and the model's inherent structural limitations, including parameter inaccuracies and improper initialization of the soil moisture conditions (Alvarado-Montero et al., 2017). Modern LSMs like the Joint UK Land Environment Simulator (JULES) require empirical soil models or pedotransfer functions (PTFs) to translate more easily measurable soil properties like soil texture into hydraulic parameters (hydraulic conductivity and soil matric suction) essential

for running the model. Errors in the specified values of PTF constants can spread through land surface models, impacting the accuracy of model forecasts.

Combining land surface models and observations synergistically can provide better soil moisture estimates than either approach in isolation. Data assimilation facilitates this process by optimally combining information from land surface models and observations measured from satellites or ground sensors, accounting for uncertainties in both the models and the observations

(Beven and Binley, 1992; Kumar et al., 2022; Xu et al., 2025). Recently, land data assimilation systems using soil moisture observations have gained considerable attention (Pinnington et al., 2018; Mason et al., 2020; Kim et al., 2021; Tian et al., 2021; Hung et al., 2022; Visweshwaran et al., 2022b; Heyvaert et al., 2024). While sequential assimilation methods (like Ensemble filters) are widely used due to their relatively simple and cost-effective implementation (Cooper et al., 2019; Gong et al., 2024), dynamically updating model states and parameters at frequent intervals may not be consistent with the stable (non-chaotic) na-

ture of land surface processes (Shi et al., 2015; De Lannoy and Reichle, 2016). On the other hand, variational techniques (like 4D-Var) can use a longer assimilation window timeframe and establish time-invariant state-parameter values over the duration of this window (Smith et al., 2013; Yang et al., 2016; Pinnington et al., 2017). Due to the high dimensionality and complexity of JULES, along with the frequent release of new versions, calculating the gradient of the dynamical model and the observation operator has become both computationally expensive and inefficient. Hybrid methods, like 4D-En-Var, approximate the model

gradient and leverage the advantages of both variational and ensemble approaches, providing more accurate estimates over long time windows without requiring the explicit calculation of the full model gradient.

Previous studies on parameter estimation in land surface models using soil moisture observations have primarily focused on directly improving the model's soil parameters to enhance prediction accuracy (Lü et al., 2011; Zha et al., 2019; Seo et al.,





2021; Visweshwaran et al., 2024). We have attempted to update the underlying PTF constants to understand their impact on model forecasts, and only a few studies have explored this on the JULES model (Pinnington et al., 2021; Cooper et al., 2021b). Addressing PTF constants is crucial, as it preserves the relationships between soil physical parameters (such as matric suction and hydraulic conductivity) and reduces uncertainty propagation from poor calibration. For studies that include multiple sites, assimilating observations from these sites together can produce a common posterior PTF constant set, representative of various soil characteristics, which can then be applied in regions lacking in-situ observations.

Beyond parameters, the model's initial soil moisture condition also plays a significant role in model forecasts. While optimizing model parameters minimizes long-term bias (systematic error) in forecasts, state assimilation addresses stochastic errors (Nearing et al., 2018). The initial soil moisture conditions influence not only current land surface processes but also future conditions through its long-term memory effect (Walker and Houser, 2001). Many studies have therefore focused on updating the initial soil moisture state in land surface models (Ridler et al., 2014; Dumedah and Walker, 2014; Chen et al., 2015).

Accurate forecasting requires precise information on both the initial conditions and model parameters (Smith et al., 2013). This dual focus on both parameter and state updates is important, yet remains underexplored in the context of the JULES land surface model. Unlike previous work, we introduce a novel approach by implementing joint state-parameter estimation for the JULES model, simultaneously updating both the PTF constants and the initial soil moisture conditions. This joint assimilation framework allows for a more integrated approach to managing uncertainty and represents the first application of this method for the JULES model, aiming to improve predictive accuracy across various soil characteristics. To demonstrate the advantage of state-parameter assimilation over individual state-only or parameter-only assimilation, in this study, we performed soil moisture assimilation in three scenarios. 1) state-only assimilation to update the initial soil moisture condition, 2) parameter-only assimilation to update the PTF constants, and 3) joint state-parameter assimilation to update both the initial soil moisture condition and Cosby's PTF constants. COSMOS-UK soil moisture observations from 2017 across 16 sites in the UK were integrated into the community JULES land surface model using a 4D-En-Var hybrid assimilation approach. Observations from all 16 sites were used together to derive common PTF constants representative of the different soil types in the UK, and the model's initial conditions for each site.

Section 2 describes the details of the study area, COSMOS-UK observations, and the JULES and Cosby models. It then outlines the methodology used in the study and the metrics applied to evaluate the model's performance before and after assimilation. Section 3 discusses the performance of JULES during state estimation, parameter estimation, and joint state-parameter assimilation, highlighting how field-scale soil moisture assimilation improves JULES's ability to estimate and forecast soil moisture. Finally, Section 4 emphasizes the significance of joint state-parameter assimilation in enhancing JULES performance across diverse soil characteristics and discusses the broader implications of these findings for future applications.





## 2    Methods

### 2.1    Study Area and COSMOS-UK Observation

To assess the performance of the JULES model before and after assimilating field-scale soil moisture observations through the 4D-En-Var scheme, 16 sites across the UK were selected (Figure 1). The sites were chosen based on soil characteristics, the availability of COSMOS-UK soil moisture, and meteorological observations. The soil texture at nine locations (Cardington: CARDT, Bickley Hall: BICKL, Crichton: CRICH, Waddesdon: WADDN, Hollin Hill: HOLLN, Easter Bush: EASTB, Rothamsted: ROTHD, Hartwood Home: HARTW, Chobham Common: CHOBH) is categorized as "typical mineral soil," where the JULES model is expected to perform well. Conversely, Sheepdrove: SHEEP, Gisburn Forest: GISBN, Moorhouse: MOORH, and Sourhope: SOURH have soils with "high organic content", while Chimney Meadows: CHIMN, Porton Down: PORTN, and Lullington Heath: LULLN feature "calcareous mineral soil". The details about the soil characteristics were taken from Antoniou et al. (2019).

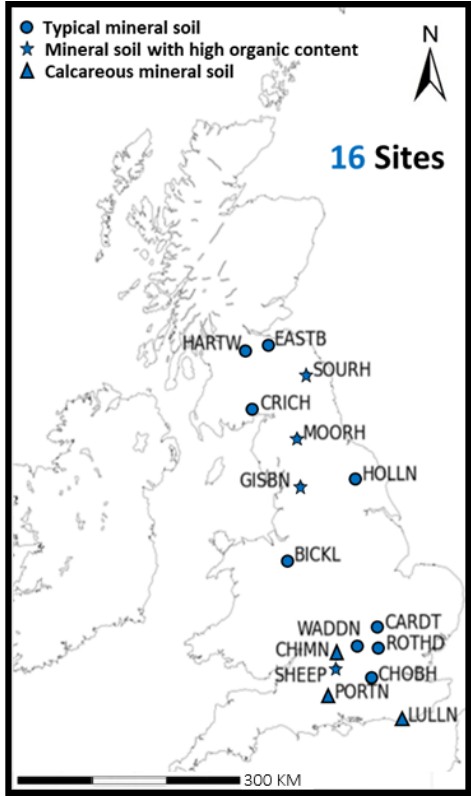

**Figure 1.** Locations of COSMOS-UK sites used for assimilation purposes.





The meteorological dataset required to run the JULES model was obtained from the COSMOS-UK sub-daily database (Cooper et al., 2021b; Stanley, 2023). These data were collected every 30 minutes at the same 16 locations as soil moisture observations were measured. Meteorological observations from 2016 to 2018 were used in this study. The year 2016 was designated as the spin-up period to initialize the JULES model, while 2017 served as the assimilation window for 4D-En-Var, and 2018 was used for forecast evaluation of JULES after assimilation.

For assimilation purposes, daily soil moisture observations were obtained from the Cosmic-ray Soil Moisture Observing System, which has ∼50 sites across the United Kingdom. This system provides soil moisture time series observations across different sites, covering different soil types, land uses, and environmental conditions. The CNRS sensor at each site monitors epithermal cosmic-ray neutrons attenuated by water molecules in the soil profile, providing near-real-time soil moisture variations integrated over a variable sensing depth; this depth is included with the dataset at a daily timestep (Evans et al., 2016). CRNS provides information for an area up to 120,000 $m^2$ and an approximate depth of 10 to 80 cm around its peripheral area, depending on local site conditions (Antoniou et al., 2019). The soil moisture estimates from the sensors are calibrated for each site to allow for local conditions (Evans et al., 2016).

## 2.2 JULES Land Surface Model

The Joint UK Land Environment Simulator is a community land surface model that can operate both independently and as the land surface component within the Met Office Unified Model (Clark et al., 2011; Best et al., 2011). Meteorological data, including precipitation, wind speed, temperature, air pressure, and solar radiation (incoming shortwave and longwave radiation), are required inputs for running the JULES model. A detailed explanation of the energy and water flux components of JULES is provided in Best et al. (2011).

The JULES model assumes four vertical soil layers to represent soil moisture conditions, with default thicknesses of 0.1 m, 0.25 m, 0.65 m, and 2 m. The Richards equation defines the vertical movement of water flux between the layers,

$$\frac{d\theta_l}{dt} = w'_{l-1} - w'_l - E'_l - R_{bl}, \tag{1}$$

where $\theta$ and the subscript $l$ represent the soil water content and soil layer, respectively, and $w'_{l-1}$ and $w'_l$ represent the downward water flux infiltrating from the layer above and out to the lower layer, respectively. The evapotranspiration uptake by the plant roots from each soil layer and the lateral movement of water are represented by $E'_l$ and $R_{bl}$, respectively. In this study, $R_{bl}$ is set to 0 since the assimilation is done separately on 16 sites, treating JULES as one-dimensional in vertical columns.

The JULES model implements Darcy's law to update the soil moisture content at each layer based on hydraulic conductivity. The Brooks and Corey model (Brooks and Corey, 1964) is used to represent the relationship between soil water content ($\theta$) and hydraulic conductivity ($K_h$),

$$\frac{\theta}{V_{\text{sat}}} = \left(\frac{\psi}{\text{sathh}}\right)^{-\frac{1}{b}}, \tag{2}$$

and

$$K_h = \text{satcon}\left(\frac{\theta}{V_{\text{sat}}}\right)^{2b+3}, \tag{3}$$





where b and $\psi$ represent the exponent in the soil hydraulic characteristic, and matric suction, respectively and $V_{\text{sat}}$, sathh, and satcon represent the volumetric soil moisture content, matric suction, and hydraulic conductivity at saturation condition, respectively. Pedotransfer functions are generally used to determine the values of these hydraulic properties, which depend on soil characteristics.

## 2.3 Cosby's Pedotransfer function

The volumetric soil moisture content can be estimated using hydraulic conductivity and soil matric suction, as proposed by the Brooks and Corey method. Pedotransfer Functions are often employed to determine the values of these parameters by establishing a relationship with easily measurable properties like soil texture and specific density (Marthews et al., 2014). In this study, the Cosby Pedotransfer Function (Cosby et al., 1984) is employed, considering the input and output data requirements of the soil hydraulic model. Furthermore, Cosby's PTF has been used in conjunction with the JULES land surface model across a wide range of UK soils and has shown promising results (e.g. Cooper et al., 2021a). The mathematical explanation of Cosby's PTF is summarized in the following equations. The soil-water retention exponent, b, is defined as,

$$\text{b} = K1 + K2 f_{\text{clay}} - K3 f_{\text{sand}}, \tag{4}$$

where $f_{\text{clay}}$, $f_{\text{silt}}$, and $f_{\text{sand}}$ represent the fraction of clay, silt and sand, respectively, in percentage. The coefficients $K1$, $K2$, and $K3$ are Cosby constants. The volumetric water content at saturation, $V_{\text{sat}}$, is represented by,

$$V_{\text{sat}} = K4 - K5 f_{\text{clay}} - K6 f_{\text{sand}}, \tag{5}$$

where $V_{\text{sat}}$ represents the maximum water-holding capacity of the soil. It depends on the soil texture via coefficients $K4$, $K5$, and $K6$. The soil matric suction (in meters), sathh, describing the force with which water is held in the soil at saturation is represented by,

$$\text{sathh} = 0.01 \times 10^{(K7 - K8 f_{\text{clay}} - K9 f_{\text{sand}})}. \tag{6}$$

Equation 6 shows a logarithmic relationship with the clay and sand fractions, and the parameters $K7$, $K8$, and $K9$ are Cosby constants. The saturated hydraulic conductivity($kg\ m^{-2}\ s^{-1}$), satcon, which describes how easily water moves through the soil when fully saturated, is represented by,

$$\text{satcon} = 10^{(-K10 - K11 f_{\text{clay}} + K12 f_{\text{sand}})}. \tag{7}$$

The critical point volumetric water content ($V_{\text{crit}}$) corresponds to the water content (in $m^3\ m^{-3}$) at a specific matric suction (usually -33 KPa, or field capacity) and the wilting point ($V_{\text{wilt}}$) is the volumetric water content (in $m^3\ m^{-3}$) at suction of -1500 KPa, beyond which plants can no longer extract water from the soil. The terms $V_{\text{crit}}$ and $V_{\text{wilt}}$ are represented by,

$$V_{\text{crit}} = V_{\text{sat}} \left( \frac{\text{sathh}}{3.364} \right)^{\frac{1}{\text{b}}}, \tag{8}$$





and

$$V_{\text{wilt}} = V_{\text{sat}} \left( \frac{\text{sathh}}{152.9} \right)^{\frac{1}{b}}.$$
(9)

Lastly, the soil thermal capacity ($h_{\text{cap}}$) representing the ability of the soil to store heat and the soil thermal conductance ($h_{\text{con}}$)
showing the measure of the soil's ability to transfer heat are represented by,

$$h_{\text{cap}} = (1 - V_{\text{sat}})(2.376 \times 10^6 f_{\text{clay}} + 2.133 \times 10^6 f_{\text{silt}}),$$
(10)

and

$$h_{\text{con}} = 0.025^{V_{\text{sat}}} \left( 1.16^{f_{\text{clay}}(1-V_{\text{sat}})} \cdot 1.57^{f_{\text{sand}}(1-V_{\text{sat}})} \cdot 1.57^{f_{\text{silt}}(1-V_{\text{sat}})} \right).$$
(11)

The hydraulic parameters b, $V_{\text{sat}}$, sathh and satcon are primary input requirements to run the JULES model and these parameters
are estimated using sand and clay fractions as shown in equations (4-7). The parameters $h_{\text{cap}}$, $h_{\text{con}}$, $V_{\text{crit}}$ and $V_{\text{wilt}}$ are derived
parameters of $V_{\text{sat}}$. In this study, we use prior values of $K1$ to $K12$ as described by Cosby et al. (1984).

## 2.4 4D-En-Var Assimilation

In this study, we adopted the 4D-En-Var hybrid assimilation method to update JULES's initial soil moisture condition and/or
Cosby PTF constants ($K1$ to $K12$). We consider the discrete non-linear dynamical model, defined as

$$\boldsymbol{z}_t = \boldsymbol{f}_{t-1 \rightarrow t} \left( \boldsymbol{z}_{t-1}, \boldsymbol{p}_{t-1} \right),$$
(12)

where $\boldsymbol{z}_t \in \mathbb{R}^n$ represents the state vector at time $t$, and $\boldsymbol{f}$ is the non-linear model operator that evolves the model state vector
from time $t-1$ to $t$ ($t = 1, 2, \ldots, N$ timesteps). The model parameter vector at time $t-1$ is described by $\boldsymbol{p}_{t-1} \in \mathbb{R}^m$, and
the parameters are assumed to be time-invariant, i.e., $\boldsymbol{p}_t = \boldsymbol{p}_{t-1}$. Initial conditions provide the starting values for $\boldsymbol{z}_0$ and are
propagated forward through the model operator.

Let $\boldsymbol{y}_t \in \mathbb{R}^{k_t}$ represent the observation vector at time $t$, where $k_t$ is the length of the observation vector. These observations
are related to the model state through the observation operator $\boldsymbol{h}_t : \mathbb{R}^n \rightarrow \mathbb{R}^{k_t}$, which maps the state vector to the observation
space:

$$\boldsymbol{y}_t = \boldsymbol{h}_t(\boldsymbol{z}_t) + \epsilon_t,$$
(13)

where $\epsilon_t \in \mathbb{R}^{k_t}$ is the observation error term, assumed to be unbiased, Gaussian, and uncorrelated in time, with a covariance
matrix $\boldsymbol{R}_t$.

Pinnington et al. (2020) developed the Land Variational Ensemble Data Assimilation Framework (LAVENDAR) for param-
eter estimation in JULES, and we modified this algorithm to update both the state and/or parameters for the three scenarios
conducted in this study. This optimization is achieved by estimating the control vector, $\boldsymbol{x}_t$, which encapsulates the variables
targeted during assimilation, such as the initial soil moisture states, PTF constants, or both, depending on the assimilation
scenario.





1. In Scenario 1 (state-only assimilation: SC 1), we aim to estimate the initial soil moisture states, denoted by $\boldsymbol{z} \in \mathbb{R}^n$, where $n = 64$ for the 4 soil layers at each of the 16 sites. The control vector for this scenario is represented as $\boldsymbol{x}_t = \begin{bmatrix} \boldsymbol{z}_t \end{bmatrix}^T \in \mathbb{R}^n$.

2. In Scenario 2 (parameter-only assimilation: SC 2), we focus on estimating the constants of the PTF, denoted by $\boldsymbol{p} \in \mathbb{R}^m$, where $m = 12$ for the 12 Cosby constants. Here, the control vector is represented as $\boldsymbol{x}_t = \begin{bmatrix} \boldsymbol{p}_t \end{bmatrix}^T \in \mathbb{R}^m$.

3. In Scenario 3 (state-parameter assimilation: SC 3), we simultaneously estimate both the initial soil moisture states and the PTF constants. The augmented system control vector $\boldsymbol{x}_t$ is then determined by appending the parameters to the model state vector. It is represented as $\boldsymbol{x}_t = \begin{bmatrix} \boldsymbol{p}_t & \boldsymbol{z}_t \end{bmatrix}^T \in \mathbb{R}^{m+n}$.

The 4D-En-Var method is similar to 4D-Var, as it assimilates all observations from a predetermined time window and spatial domain together in a batch. In 4D-Var, we seek to minimize the cost function ($\boldsymbol{J}$), which is the sum of the discrepancy between the model estimate and the prior term ($\boldsymbol{x}_b$ representing the prior vector at time = 0, 'b' for 'background' synonymous with 'prior'), weighted by the background error covariance matrix ($\boldsymbol{B}$), and the differences between all the model states and observations, weighted by the observation error covariance matrix ($\boldsymbol{R}$). The cost function ($\boldsymbol{J}$) is defined as

$$\boldsymbol{J}(\boldsymbol{x}_a) = \frac{1}{2}(\boldsymbol{x}_a - \boldsymbol{x}_b)^T \boldsymbol{B}^{-1}(\boldsymbol{x}_a - \boldsymbol{x}_b) + \frac{1}{2}\sum_{t=0}^{N}(\boldsymbol{h}_t(\boldsymbol{x}_t) - \boldsymbol{y}_t)^T \boldsymbol{R}^{-1}(\boldsymbol{h}_t(\boldsymbol{x}_t) - \boldsymbol{y}_t), \tag{14}$$

where N represents the length of the assimilation window. The principal aim is to optimize a state and/or parameter vector (posterior vector: $\boldsymbol{x}_a$, 'a' for 'analysis' synonymous with 'posterior') that minimizes $\boldsymbol{J}$. The posterior vector is usually determined by applying a gradient-based descent algorithm iteratively to the cost function and its gradient. To avoid calculating tangent linear model, adjoint model and $\boldsymbol{B}^{-1}$ for JULES, 4D-En-Var applies a control variable transform to precondition the matrix $\boldsymbol{B}$ (Bannister, 2017). This approach allows $\boldsymbol{B}$ to be defined as the product of the perturbation matrix, $\boldsymbol{x}'_b$, and its transpose given by

$$\boldsymbol{B} \approx \boldsymbol{x}'_b \boldsymbol{x}'^T_b, \tag{15}$$

where

$$\boldsymbol{x}'_b = \frac{1}{\sqrt{N_e - 1}}\left(\boldsymbol{x}^1_b - \bar{\boldsymbol{x}}_b, \boldsymbol{x}^2_b - \bar{\boldsymbol{x}}_b, \ldots, \boldsymbol{x}^{m-1}_b - \bar{\boldsymbol{x}}_b, \boldsymbol{x}^{N_e}_b - \bar{\boldsymbol{x}}_b\right). \tag{16}$$

The posterior vector $\boldsymbol{x}_a$ is then defined as

$$\boldsymbol{x}_a = \bar{\boldsymbol{x}}_b + \boldsymbol{x}'_b \boldsymbol{w}, \tag{17}$$

where $\boldsymbol{w}$ is a vector of length $N_e$ and $N_e$ is the ensemble size. The term $\boldsymbol{x}^i_b$ can come from a previous forecast (in that case $\bar{\boldsymbol{x}}_b$ is the mean of the $N_e$ ensemble members) or sampled from a known distribution $\mathcal{N}(\bar{\boldsymbol{x}}_b, \boldsymbol{B})$. Substituting equations 17 and 15 into 14, we get a new cost function and its gradient as a function of $w$ given by

$$\boldsymbol{J}(\boldsymbol{w}) = \frac{1}{2}\boldsymbol{w}^T \boldsymbol{w} + \frac{1}{2}\sum_{t=0}^{N}\left(\boldsymbol{y}'_{b,t}\boldsymbol{w} + \boldsymbol{h}_t(\bar{\boldsymbol{x}}_b) - \boldsymbol{y}_t\right)^T \boldsymbol{R}^{-1}\left(\boldsymbol{y}'_{b,t}\boldsymbol{w} + \boldsymbol{h}_t(\bar{\boldsymbol{x}}_b) - \boldsymbol{y}_t\right), \tag{18}$$



and

$$220 \quad \nabla \boldsymbol{J}(\boldsymbol{w}) = \boldsymbol{w} + \sum_{t=0}^{N} \left(\boldsymbol{y}'_{b,t}\right)^T \boldsymbol{R}^{-1} \left(\boldsymbol{y}'_{b,t}\boldsymbol{w} + \hat{\boldsymbol{h}}_t(\bar{\boldsymbol{x}}_b) - \hat{\boldsymbol{y}}_t\right). \tag{19}$$

The term $\sum_{t=0}^{N} \boldsymbol{y}'_{b,t} \approx \sum_{t=0}^{N} \boldsymbol{h}_t(\boldsymbol{x}'_b)$ is the perturbation matrix in observation space, and it is computed by

$$\boldsymbol{y}'_{b,t} = \frac{1}{\sqrt{N_e - 1}} \left(\boldsymbol{h}_t\left(\boldsymbol{x}_b^1\right) - \boldsymbol{h}_t\left(\bar{\boldsymbol{x}}_b\right), \boldsymbol{h}_t\left(\boldsymbol{x}_b^2\right) - \boldsymbol{h}_t\left(\bar{\boldsymbol{x}}_b\right), \dots, \boldsymbol{h}_t\left(\boldsymbol{x}_b^{N_e-1}\right) - \boldsymbol{h}_t\left(\bar{\boldsymbol{x}}_b\right), \boldsymbol{h}_t\left(\boldsymbol{x}_b^{N_e}\right) - \boldsymbol{h}_t\left(\bar{\boldsymbol{x}}_b\right)\right). \tag{20}$$

This allows the computation of the posterior without explicitly determining the tangent linear model, adjoint model, and $\boldsymbol{B}^{-1}$.
Minimizing the cost function (equation 18) using the gradient provides the posterior estimate of $\boldsymbol{w}$, which, when substituted
into equation 17, yields the posterior estimate of the control vector $\boldsymbol{x}_a$.

## 2.5 Experiment Design

The implementation of COSMOS-UK soil moisture assimilation into the JULES model using the 4D-En-Var is done in three
scenarios (as outlined in Section 2.4). In the first and second scenarios, the initial soil moisture state and Cosby's PTF constants
are updated individually, respectively, to understand their individual impact on the JULES forecast. In the third scenario, joint
state-parameter assimilation is performed to understand the combined impact. The steps involved in the implementation are
outlined below and shown in Figure 2.

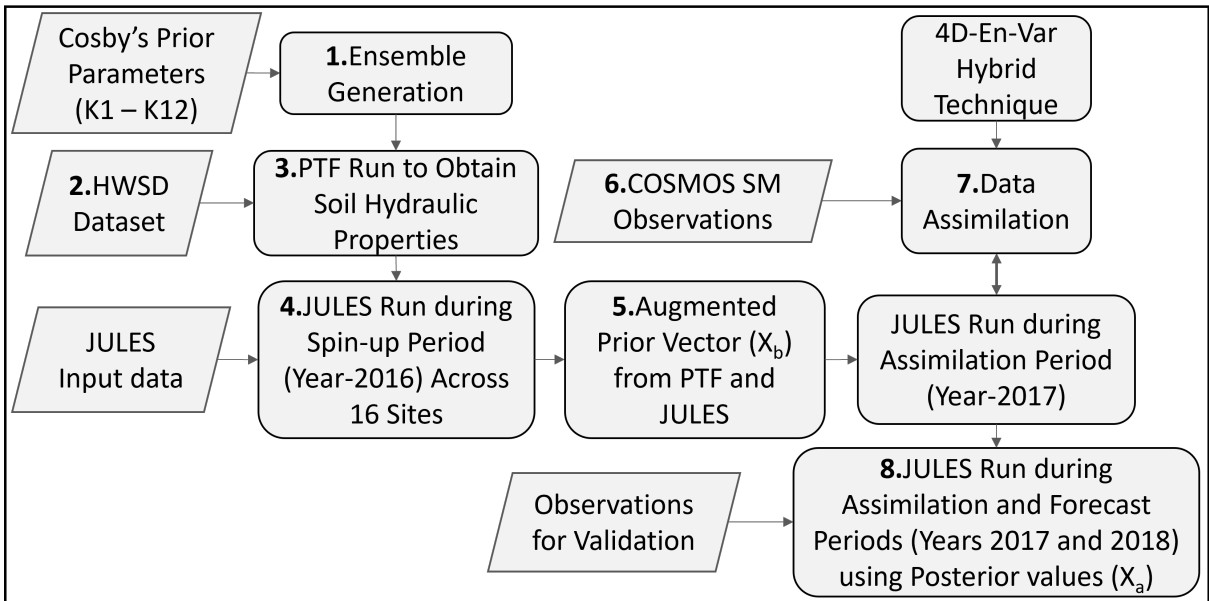

**Figure 2.** Schematic showing the application of 4D-En-Var assimilation scheme on JULES land surface model during state-parameter
estimation (Scenario 3).The numbered boxes represent the steps adopted in this study.





1. **Ensemble generation:** A fifty-member ensemble (i.e. $N_e = 50$) was generated for each of the 12 Cosby PTF constants by sampling from a Gaussian distribution with mean values as outlined in Cosby et al. (1984). A 10 percent standard deviation around the mean was used, following the successful implementation of JULES on these sites by Cooper et al. (2021a).

2. **Harmonized World Soil Database (HWSD):** Soil texture information, comprising the percentage of sand, silt and clay, is obtained from the HWSD dataset (Fischer et al., 2008) for all 16 sites. The HWSD data, originally in a 30 arc-second spatial scale gridded raster format, was downscaled to a 1 km resolution, and the gridpoints passing through the COSMOS-UK sites were selected.

3. **Pedotransfer function:** Cosby's PTF is applied using the HWSD data and the 50 $K1 - K12$ sets to generate a 50-member ensemble of unique soil hydraulic parameters.

4. **Spin-up period:** The soil hydraulic parameters ($50 \times 12$) obtained from the PTF are fed into the JULES model during the warm-up period extending from January to December 2016.

5. **Prior vector:**

   (a) **Scenario 1 (SC 1):** During state estimation, the JULES 50-member ensemble forecast at the end of the spin-up period, represented by four soil layers, forms the initial soil moisture condition for the start of the assimilation period. These 4 layers for each of the 16 sites are combined to represent the prior control vector ensemble (i.e., $\{\boldsymbol{x}_b^{(i)} \in \mathbb{R}^{64} : i = 1, 2, \ldots, N_e\}$).

   (b) **Scenario 2 (SC 2):** During parameter estimation, Cosby's prior parameter 50-member ensemble, which is common to all 16 sites, forms the prior ensemble with a dimension of $\{\boldsymbol{x}_b^{(i)} \in \mathbb{R}^{12} : i = 1, 2, \ldots, N_e\}$.

   (c) **Scenario 3 (SC 3):** During state-parameter estimation, JULES 50-member ensemble forecast (containing 4 soil layers) from the spin-up period for each site and a common Cosby's prior parameter ensemble representing all the 16 sites are augmented to represent the prior ensemble (i.e. $\{\boldsymbol{x}_b^{(i)} \in \mathbb{R}^{76} : i = 1, 2, \ldots, N_e\}$).

6. **Observations:** As outlined in Section 2.1, COSMOS-UK provides soil moisture observations at field scale around the CRNS instrument at a 30-minute frequency. We require the JULES soil moisture values corresponds to the same soil depth as the observations. Therefore, a weighted depth approach is employed, averaging the JULES estimates (soil moisture over 4 layers) to adjust to the level of COSMOS-UK soil moisture depth. It should be noted that hourly observations are averaged to a daily scale, with analysis indicating that the standard deviation of the hourly data around the daily mean is approximately 20% (Cooper et al., 2021a). Further, observation errors are assumed to be uncorrelated. Considering these aspects, a standard deviation of 50% of the daily mean observation is assigned to $\boldsymbol{R}$ for these sites. Further details on the weighted depth approach and the choice of $\boldsymbol{R}$ can be found in Cooper et al. (2021a).

7. **Data assimilation:** The 4D-En-Var assimilation scheme is employed within the JULES model for the year 2017 (1-year time window) to obtain posterior values for the state (for each site) and/or parameters (common for all sites).





8. **Model forecast:** Steps 3 and 4 are repeated with posterior PTF values, and the JULES model is rerun for the assimilation period (year 2017) and for the forecast period (year 2018) with the posterior initial conditions and soil ancillary files.

## 2.6 Evaluation Metrics

To assess the model's performance during assimilation and forecast periods, two evaluation metrics are calculated: Root Mean Square Error (RMSE) and Kling-Gupta Efficiency (KGE). These metrics enable a comprehensive assessment of the model's performance by quantifying discrepancies between observed and modelled soil moisture values over a two-year period. The RMSE and KGE metrics are defined as follows,

$$\text{RMSE} = \sqrt{\frac{\sum_{t=1}^{N} \left( \text{SM}_\text{m}^t - \text{SM}_\text{o}^t \right)^2}{N}}, \tag{21}$$

where $N$ is the total number of observations. In this study, N = 730 for the two-year study period, and $\text{SM}_\text{m}$ and $\text{SM}_\text{o}$ are the modelled and observed soil moisture values, respectively. The term KGE is defined as

$$\text{KGE} = 1 - \sqrt{(r-1)^2 + (\alpha-1)^2 + (\beta-1)^2}. \tag{22}$$

The KGE, as proposed by Gupta et al. (2009), provides an overall assessment of soil moisture distribution trends by incorporating three components: correlation, variability, and bias. In this context, $r$ represents the Pearson correlation coefficient, capturing the correlation between modelled and observed soil moisture time series. The term $\alpha = \sigma_m / \sigma_o$ reflects the model's ability to capture the variability in the observations, where $\sigma_m$ and $\sigma_o$ are the standard deviations of the modelled and observed data, respectively. Meanwhile, $\beta = \mu_m / \mu_o$ represents the model bias, where $\mu_m$ and $\mu_o$ denote the mean soil moisture values for the model and observations. Together, these components provide a robust evaluation of model performance.

For all components and the KGE, values closer to 1 indicate good model performance. In general, the KGE value ranges from $-\infty$ to 1, with 1 representing a perfect fit. A KGE value of 0 signifies model performance equal to the average value of the observations, and any value below -0.14 indicates poor model performance (Knoben et al., 2019).

## 3 Results

### 3.1 Scenario 1: Individual effect of state assimilation on JULES performance

During state assimilation, the initial conditions are updated for each site, and the posterior JULES estimates are compared against COSMOS-UK observations. The adjustments made to the initial conditions during assimilation shifted the posterior estimates closer to the observations during the initial phase, as illustrated in Figure 3 with examples from the HOLLN and SHEEP sites. These sites were chosen to represent two distinct soil types—typical mineral soil (HOLLN) and soil with high organic content (SHEEP) since JULES performance is likely to vary depending on soil characteristics. Although the initial adjustments improved the estimates, this effect decayed rapidly after 3 to 4 months. Notably, for most sites, including HOLLN and SHEEP, both the prior and posterior estimates showed no significant differences after 6 months. This can be attributed to




the dynamic nature of soil moisture, which diminishes the impact of updated initial conditions on JULES estimates over longer durations (seasonal to annual scale).

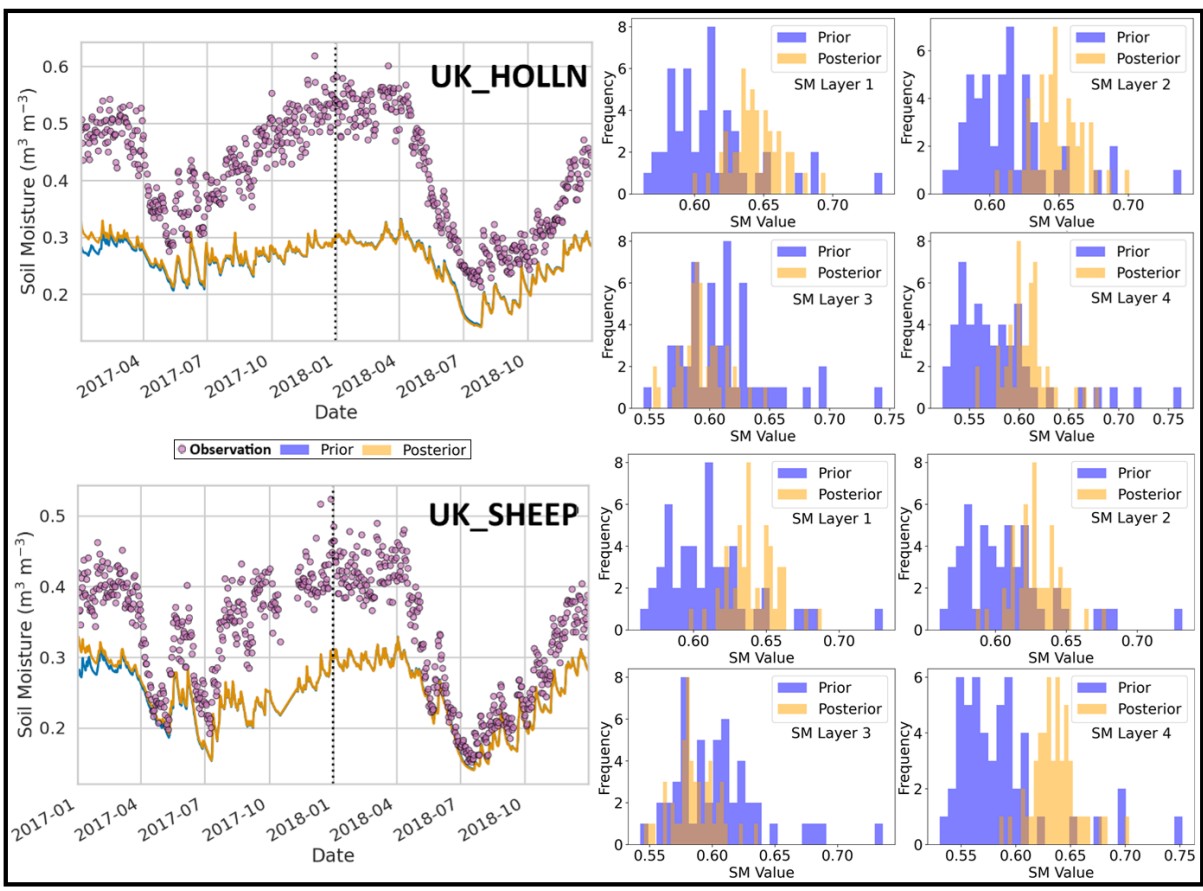

**Figure 3.** The figure shows the results of the state-only assimilation scenario (SC 1). The left panels display the modeled (ensemble mean) and observed soil moisture time series at the HOLLN and SHEEP sites during the assimilation (2017) and forecast (2018) periods. The right panels show the prior and posterior distributions of the initial condition state (January 1, 2017) across 50 ensemble members and four soil layers.

KGE metrics (Figure 4) also showed a similar outcome, with moderate improvements between the prior (Blue) and posterior (Brown) values, except at the GISBN site. Although Figure 4 includes results from other scenarios as well, this section specifically discusses scenario 1. The anomaly at the GISBN site might be due to the presence of a high amount of coniferous trees, which can complicate the calibration of the CRNS instrument and reduce the reliability of COSMOS-UK soil moisture observations. Across all sites, the posterior JULES model showed improvement in the $\alpha$ and $\beta$ components (capturing the variability and bias, respectively). However, there were no significant changes in the correlation component ($r$) of the model. This
is due to the high correlation between prior values and observations before assimilation. Table 1 shows the RMSE between the





modelled soil moisture and COSMOS-UK observations. For the state-only scenario, RMSE values reduced at all sites, with an average reduction of 39.5% for all 16 sites.

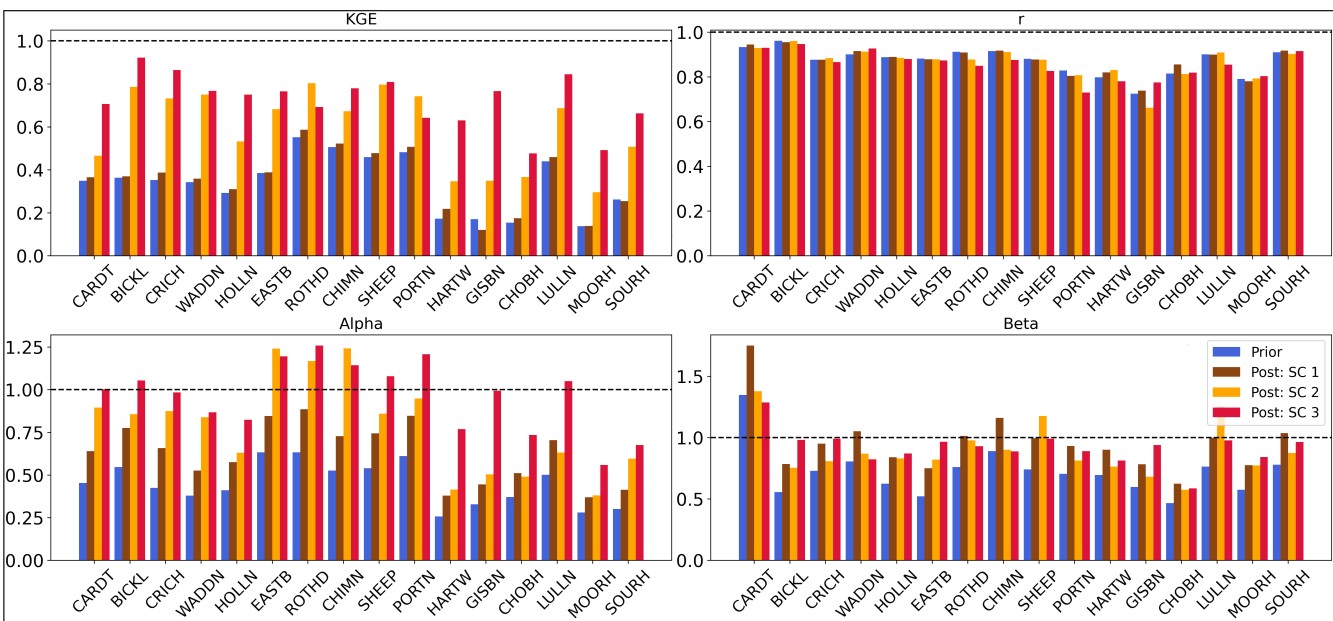

**Figure 4.** KGE metrics and its components showing the performance of JULES model across all the 16 sites in UK for three assimilation scenarios: state-only (SC 1), parameter-only (SC 2), and joint state-parameter (SC 3).




**Table 1.** RMSE values between the JULES simulated soil moisture and COSMOS-UK observations.

| Site | RMSE | | | |
|---|---|---|---|---|
| | Prior | Posterior state-only (SC 1) | Posterior parameter-only (SC 2) | Posterior state-parameter (SC 3) |
| CARDT | 0.09 | 0.07 | 0.08 | 0.08 |
| BICKL | 0.14 | 0.08 | 0.03 | 0.03 |
| CRICH | 0.12 | 0.06 | 0.04 | 0.04 |
| WADDN | 0.10 | 0.06 | 0.07 | 0.07 |
| HOLLN | 0.17 | 0.10 | 0.06 | 0.06 |
| EASTB | 0.16 | 0.10 | 0.04 | 0.03 |
| ROTHD | 0.08 | 0.03 | 0.04 | 0.04 |
| CHIMN | 0.06 | 0.06 | 0.05 | 0.05 |
| SHEEP | 0.10 | 0.05 | 0.05 | 0.05 |
| PORTN | 0.11 | 0.06 | 0.07 | 0.06 |
| HARTW | 0.17 | 0.10 | 0.10 | 0.10 |
| GISBN | 0.23 | 0.17 | 0.09 | 0.07 |
| CHOBH | 0.28 | 0.21 | 0.20 | 0.20 |
| LULLN | 0.09 | 0.04 | 0.04 | 0.04 |
| MOORH | 0.25 | 0.17 | 0.13 | 0.11 |
| SOURH | 0.13 | 0.08 | 0.07 | 0.06 |

### 3.2 Scenario 2: Individual effect of parameter assimilation on JULES performance

During parameter-only assimilation, Cosby's PTF constants were updated, leading to shifts in the posterior PTF constants compared to their respective prior values, as shown in Figure 5. This shift resulted in changes to the model's hydraulic parameters, illustrated in Figure 6. For instance, the parameter $V_{\mathrm{sat}}$ increased due to higher values of $K4$ and $K5$, suggesting an increase in the model's representation of the soil's water-holding capacity at full saturation. In contrast, satcon decreased due to changes in $K10$ to $K12$ values. This indicates that the model now simulates slower water movement through the soil when fully saturated,

which may suggest less permeable soil conditions in the model. In addition, parameters sathh and b were reduced due to the decreased values of $K7$ to $K9$ and changes in $K1$ to $K3$ values, respectively. Finally, $V_{\mathrm{crit}}$ and $V_{\mathrm{wilt}}$ did not exhibit significant changes between prior and posterior distributions. Overall, the posterior distributions are narrower than the prior distributions, since the 4D-En-Var technique serves as an approximate minimum variance estimator.





**Figure 5.** Histograms showing the prior and posterior distribution values of Cosby's PTF constants ($K1$ to $K12$) during parameter-only assimilation scenario (SC 2) across 16 sites. Blue (prior) and orange (posterior) lines represent kernel density estimates. Bin widths are individually adjusted to reflect the range and spread of values for each constant.





**Figure 6.** Ensemble prior and posterior values of different soil hydraulic parameters at 16 sites across the UK during parameter-only assimilation scenario (SC 2).

Figure 7 shows the ensemble mean JULES and COSMOS-UK soil moisture time series for the HOLLN and SHEEP sites,
which have different soil characteristics. From the figure, it is evident that the posterior soil moisture time series is more consistent with the COSMOS-UK observations than the prior values. Visually, the model estimates exhibit better correlation and lower bias with the COSMOS-UK observations. KGE metrics and their components also showed a similar outcome, indicating improved performance for the posterior estimates (Figure 4). The reduction in bias was a major contributor to the improvement in KGE metrics. This is due to changes in the $V_{\text{sat}}$, b, and satcon values, which allowed more water to be stored in
the modelled soil, thereby reducing the underestimation in JULES soil moisture estimates. When we analyzed the differences in RMSE values (Table 1) between the prior and parameter-only scenarios, the posterior RMSE values showed decreased values, with an average reduction of $47.4\%$ across all 16 sites.





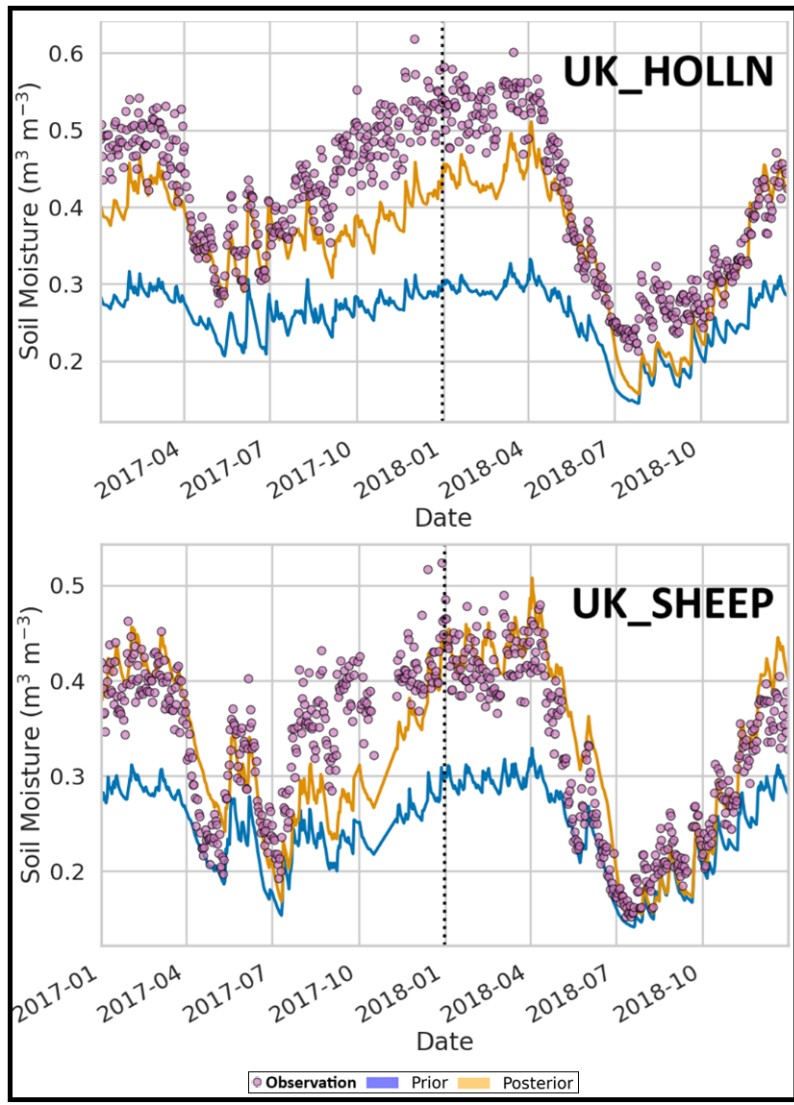

**Figure 7.** Modelled (ensemble mean) and observed soil moisture time series at HOLLN and SHEEP sites during assimilation (2017) and forecast periods (2018) for parameter-only assimilation scenario (SC 2).

## 3.3    Scenario 3: Combined effect of state-parameter assimilation on JULES performance

During joint state-parameter assimilation, we optimized both Cosby's constants (Figure 8) and JULES's initial soil moisture
condition (Figure 10). Figure 8 shows that the posterior parameter distribution (orange) deviated from its respective prior
distribution (blue) after assimilation, though the deviations are less pronounced compared to the parameter-only scenario. This
suggests that updating the initial soil moisture conditions in this scenario has partially alleviated the adjustments required
in the PTF constants. Due to this, all eight soil hydraulic properties in the Brooks and Corey model also showed deviations





from their respective prior values as shown in Figure 9. In general, the ensemble members span a narrower range in the
posterior distribution than in the prior distribution, as seen in the parameter-only assimilation scenario, suggesting a reduction
in variability of the hydraulic parameters after assimilation. The values of satcon were reduced due to the decreased values of
$K10$ and $K12$. Conversely, $V_{\text{sat}}$ showed an increased value due to higher $K4$ and $K5$ values after assimilation. However, both
$V_{\text{sat}}$ and satcon exhibited smaller changes compared to the parameter-only scenario. Finally, b and sathh did not show much
change in distribution after assimilation. These results indicate that the posterior JULES can store more water in the soil due
to the combined effect of higher retention capacity and slower movement of water through the soil.

**Figure 8.** Histograms showing the prior and posterior distribution values of Cosby's PTF constants ($K1$ to $K12$) during state-parameter
assimilation (SC 3) across 16 sites. Blue (prior) and orange (posterior) lines represent kernel density estimates. Bin widths are individually
adjusted to reflect the range and spread of values for each constant.

Similarly, Figure 10 shows the prior and posterior distribution of initial soil moisture conditions for two locations, GISBN
and CARDT. These sites were selected to illustrate the assimilation's impact on different soil types— GISBN with high organic





content and CARDT with mineral-rich soil. From the figure, we can see that the initial soil moisture condition has increased in GISBN and decreased in CARDT after assimilation. This is because, on January 1, 2017, the model's prior estimates were

lower compared to the observations for the GISBN location and higher for the CARDT location. Accordingly, the initial condition was updated to be closer to the observation values. The prior and posterior distributions for all 16 locations are shown in Figures A1- A4 of Appendix A.

**Figure 9.** Ensemble prior and posterior values of different soil hydraulic parameters at 16 sites across the UK during state-parameter assimilation scenario (SC 3).

Figure 11 shows the ensemble mean soil moisture time series for two locations with different soil characteristics. Table 1 and Figure 11 show that the posterior soil moisture estimates are more consistent with the COSMOS-UK observations. RMSE

reduced by an average of $50\%$ across all sites, with the maximum reduction observed at BICKL ($78.5\%$) followed by GISBN ($69.5\%$). Overall, these results suggest that the bias in the model is significantly reduced, the posterior JULES estimates showed a good correlation with COSMOS-UK observations, and the variability in the observations is captured more accurately.



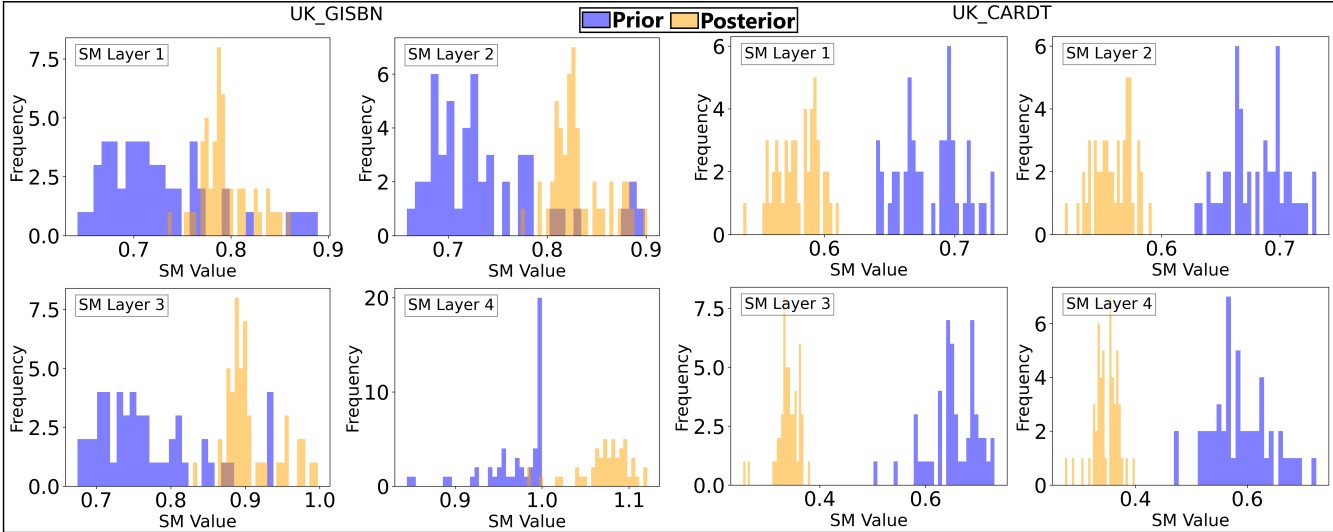

**Figure 10.** Prior (blue) and posterior (orange) distributions of JULES's initial soil moisture condition during state-parameter assimilation (SC 3) for two sites (GISBN and CARDT) across 50 ensemble members and four soil layers. Bin widths vary across subplots to allow comparison of prior and posterior value ranges for each soil layer.

KGE values showed significant improvement after assimilation, with the maximum increase observed at the GISBN site ($\approx 350\%$) and a minimum increase of $\approx 25\%$ at the CHIMN site (Figure 4). Breaking down the components of KGE, the combined effect of the $\alpha$ and $\beta$ components contributed substantially to the increase in KGE. The $r$ component was already high in the prior JULES estimates (averaging more than $\approx 0.8$), leaving little scope for assimilation to improve the correlation with COSMOS-UK observations.

Out of the 16 locations, prior soil moisture estimates at 15 sites were negatively biased, as indicated by the $\beta$ component (values less than 1, represented by a horizontal line). After assimilation, the hydraulic parameter satcon decreased, reducing the movement of water within the soil profile, while the parameter $V_{\text{sat}}$ increased, improving the water holding capacity of the soil. This combined effect increased the water content in the soil column, reducing the underestimation of the prior JULES run. This improvement also enabled the model to better capture the variability in COSMOS-UK observations, leading to an improved $\alpha$ component.



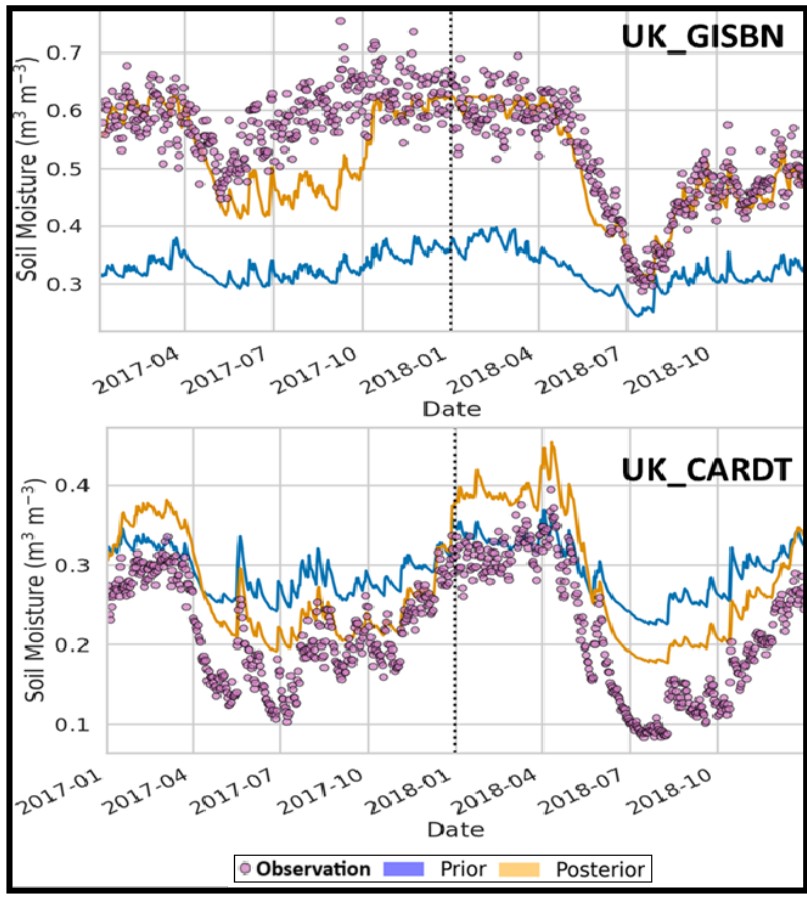

**Figure 11.** Modelled (ensemble mean) and observed soil moisture at GISBN and CARDT sites during assimilation (2017) and forecast periods (2018) for state-parameter assimilation scenario (SC 3).

## 4 Conclusions

In this work, we assessed the impact of assimilating field-scale soil moisture observations on the JULES community land surface model. We used daily COSMOS-UK observations for the year 2017 across 16 sites in the UK, employing the 4D-En-Var hybrid assimilation technique to update the state and/or parameter prior vector. While past studies focused on updating the initial condition of JULES (Seo et al., 2021) or by updating the underlying PTF functions (Pinnington et al., 2021; Cooper et al., 2021a), we performed state-parameter assimilation by simultaneously optimizing both the constants of the Pedotransfer

function and JULES initial soil moisture condition.

To demonstrate the advantage of joint state-parameter assimilation over individual state-only or parameter-only assimilation, we designed three assimilation scenarios. Scenarios 1 and 2 determined the impact of optimizing initial soil moisture conditions and Cosby's PTF constants, respectively. In scenario 3, we assessed the impact of jointly optimizing both Cosby's PTF





constants and initial condition by performing joint state-parameter assimilation.

In general, results from all the experiments showed that it is possible to improve the performance of JULES by assimilating field-scale soil moisture observations. Comparing the results of Scenario 3 with scenarios 1 and 2 demonstrated that state-parameter assimilation can constrain the JULES model to perform much better. During Scenario 3, the average KGE value across 16 sites increased from 0.33 to 0.72 after assimilation, representing an increase of 143%. In contrast, individual state-only and parameter-only assimilation results showed only moderate improvements of 7.6% (KGE: 0.33 to 0.35) and 100%

(KGE: 0.33 to 0.66), respectively.

    Soil characteristics have a strong influence on JULES performance. In general, the JULES model performs well in "typical mineral soil" or "calcareous soil" but tends to show more errors in areas with high organic content. This may be due to the incorrect initialization of PTF constants and initial conditions during the warm-up period. Prior model results (Figure 4) demonstrate this pattern, with an average KGE of 0.35 for "typical mineral soil" sites and 0.23 for "high organic content" sites. However,

assimilation results (scenario 3) showed maximum improvement at sites where JULES initially performed poorly. For instance, if we rank the model improvement across the 16 sites, three of the top four improvements were seen at sites with high organic content. GISBN, CHOBH, and MOORH showed improvements of approximately 350%, 250%, and 220%, respectively, in KGE values. The post-assimilation model with updated PTF constants and initial conditions performed well across various soil types within the UK, suggesting that this framework has strong potential for broader application within similar regions.

During state-only assimilation, the posterior JULES model performed marginally better than the prior run, with a mean improvement of 7.6% in KGE and 39.5% in RMSE. However, the effect of the updated initial soil moisture condition decayed quickly in less than six months. For a few sites, the prior and posterior soil moisture time series coincided within three months from the point of assimilation. This limitation can be addressed by assimilating more frequently using a smaller time window. Having a longer time window of one year for parameter estimation and multiple shorter time windows of three months for state

assimilation could potentially yield better forecasts.

    In conclusion, this study presents the first attempt at joint state-parameter assimilation on the JULES model, leading to significant improvements in soil moisture estimates. By optimizing both the initial conditions and PTF constants, we demonstrated clear enhancements in model performance. These improvements have important implications for better flood forecasting and disaster management, as more accurate soil moisture predictions can enhance preparedness and response strategies. This work

provides a valuable basis for further advancements in land surface data assimilation across different regions and soil types.

*Code and data availability.* JULES source code and the assimilation code used in this study are available from the MetOffice JULES repository (https://code.metoffice.gov.uk/trac/jules, last access: 30 August 2024) under Rose suite number u-dc794. Registration is required. COSMOS-UK data are deposited annually in the NERC Environmental Information Data Centre (EIDC) (https://doi.org/10.5285/b5c190e4-e35d-40ea-8fbe-598da03a1185, Stanley et al., 2021).



*Author contributions.*    VR, EC, and SD devised the experiments.VR designed the Rose suite used here and ran the experiments. EC prepared
the COSMOS-UK data. SD supervised the work and VR prepared the manuscript with inputs from all the co-authors.

*Competing interests.*    The authors declare that they have no conflict of interest.

*Acknowledgements.*    This work was jointly supported by the Natural Environment Research Council's (NERC) HydroJULES-NEXT project
(grant number: NE/X019063/1) and the Climate Change in the Arctic and North Atlantic Region and Impacts on the UK (CANARI) project
(grant number: NE/W004984/1) as a part of the National capability funding programmes. The authors gratefully acknowledge the provision
by UKCEH of hydrometeorological and soil data collected by the COSMOS-UK project. COSMOS-UK is funded by the NERC (award no.
NE/R016429/1) as part of the UK-SCAPE programme.



## Appendix A: Updated JULES initial soil moisture condition during state-parameter scenario

**Figure A1.** Prior and posterior distribution values of JULES's initial soil moisture condition at CARDT, BICKL, CRICH, and WADDN COSMOS sites during state-parameter assimilation.





**Figure A2.** Prior and posterior distribution values of JULES's initial soil moisture condition at HOLLN, EASTB, ROTHD, and CHIMN COSMOS sites during state-parameter assimilation.







**Figure A3.** Prior and posterior distribution values of JULES's initial soil moisture condition at SHEEP, PORTN, HARTW, and GISBN COSMOS sites during state-parameter assimilation.



**Figure A4.** Prior and posterior distribution values of JULES's initial soil moisture condition at CHOBH, LULLN, MOORH, and SOURH COSMOS sites during state-parameter assimilation.



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
