# Peer review of "Improving JULES Soil Moisture Estimates through 4D-En-Var Hybrid Assimilation of COSMOS-UK Soil Moisture Observations"

_EGUsphere, 2024_

## Author Comment (AC1)

**Improving JULES Soil Moisture Estimates through 4D-En-Var Hybrid Assimilation of COSMOS-UK Soil Moisture Observations**

**Response to Anonymous Referee #1 – Visweshwaran et. al (2025) doi:10.5194/egusphere-2024-3980**

The authors thank the reviewer for their insightful comments. Below we list each comment and have numbered them for ease of reference. Our responses are provided below in purple. References are as cited in the manuscript or provided in each response.

1) A big part of the methodology and the monitoring data set have been presented already in Cooper et al., 2021a (https://doi.org/10.5194/hess-25-2445-2021). The second scenario of the manuscript seems to be very similar to this work published by the manuscript's second author as main author. It is using the same land surface model JULES, the same pedotransfer functions including the same starting parameter values, the same 16 cosmic-ray neutron locations out of the about 50 sites of the COSMOS-UK network, the same year for the data assimilation and the same following year for the forecast, and the same ensemble size. The 4D-En-Var assimilation method outlined in the manuscript is depicted already in Pinnington et al. (2020), also including a combination of parameter vector and state vector, and seems to be part of LAVENDAR already together with JULES.

Thank you for this comment. Pinnington et al. (2020) developed the LAVENDAR framework and later applied for **parameter estimation** in JULES for Toth PTF constants (Pinnington et al., 2021). Cooper et al. (2021a) applied this framework for **parameter estimation** to optimize Cosby pedotransfer function (PTF) constants. However, their implementation focused solely on parameter estimation and **did not include any state estimation**. Here we introduce a novel extension by incorporating **joint state-parameter estimation** by modifying the algorithm to incorporate the state vector alongside the parameter vector in an augmented approach, enabling a more comprehensive data assimilation scheme. This dynamically adjusts both soil moisture initial conditions and soil hydraulic parameters (through PTF constants) within the 4D-En-Var assimilation framework.

To assess the impact of simultaneously updating both states and parameters (Scenario 3 in our results) and to understand how the model responds to this approach, it is necessary to compare it against state-only (Scenario 1) and parameter-only (Scenario 2) estimation. We acknowledge that this distinction was not clearly articulated in the introduction and conclusion sections and we will revise these.

2) While the study presented in the manuscript does go somewhat further by now also applying the data assimilation method to include the initial soil moisture state, it does not describe the methods and results of the manuscript on basis of the existing work but as an unclear mix with vague formulations of origin. These publications are cited but often rather as context. A clear distinction between existing work and own work is necessary. Maybe there are reasons to use exactly the same setting as before, though using another data set and approach could contribute more novelty, but this needs to be discussed and rectified. Also results should be discussed on basis of existing work, not in a diffuse way. To give one simple example, in the conclusions, second paragraph, the manuscript presents that KGE has improved from 0.33 to 0.66 for the parameter-only assimilation, which is identical to the statement in the conclusion of Cooper et al. (2021a) that 'we see an

improvement in the average KGE metric from 0.33 (range 0.10 to 0.69) before data assimilation to an average of 0.66 after data assimilation'. It should be referred to the previous result and made clear that this finding is identical to this previous result and why or to be discussed how it nevertheless may be different and why.

As noted in our response to Comment 1, we will revise the introduction and conclusion sections to clarify the distinction between this study and previous work such as Cooper et al. (2021a). To assess the impact of simultaneously updating both state and parameter estimates (Scenario 3) and to enable a direct comparison with state-only (Scenario 1) and parameter-only (Scenario 2) assimilation, we deliberately use the same observation data (COSMOS-UK in-situ observations), assimilation period (2017), forecast period (2018), land surface model (JULES), PTF selection (Cosby). The prior values of the PTF constants follow Cosby et al. (1984) and Marthews et al. (2014), consistent with previous applications in JULES and other land surface models. To ensure that all three scenarios in this study are run using exactly the same model configuration and processing, we repeat the parameter-only assimilation experiment rather than relying on results from Cooper et al. (2021a). Re-running the experiment avoids the risk of introducing uncertainty due to possible differences in implementation or processing steps.

In addition, for the benefit of readers, we include Scenario 2 in this work so that the comparison across all three scenarios can be followed within a single study, without requiring reference to previous work. After performing the experiment, we observe that the results for Scenario 2, including KGE and RMSE, are identical to those reported in Cooper et al. (2021a), and this will be clearly stated and referenced in the revised manuscript.

3) Features of cosmic-ray neutron sensing are partly wrong and even contradictory within the manuscript. For example, in line 30 the horizontal footprint size is specified as 'approximately 25 to 30 hectares', in line 112 then 'an area up to 120,000 m$^2$', which is only half of the former.

Thank you for pointing out this inconsistency. The horizontal footprint size of the cosmic-ray neutron sensing (CRNS) instrument spans approximately 12 hectares (i.e., 120,000 m²). We will revise line 30 accordingly.

4) Also, the depth specifics are not explained adequately and the fourth, deepest layer of the JULES model actually is not linked to the soil moisture observation by cosmic-ray neutron sensing at all, and the third layer likely only sometimes. And, the equations used for a weighted average of model layer soil moisture values to compare to the observed soil moisture (Cooper et al., 2021a, Pinnington et al., 2021) is a mere average accounting for the different layer thicknesses but not accounting for the strongly decaying weight with depth of cosmic-ray neutron sensing.

We will clarify the depth ranges of the JULES soil layers in the revised manuscript. While we acknowledge that the weighted averaging operator used does not explicitly account for the depth-dependent sensitivity of cosmic-ray neutron sensing, previous studies (Cooper et al., 2021a; Pinnington et al., 2021) have shown this operator to be effective in practice.

We have also tested more complex operators and found that the data assimilation results were not particularly sensitive to the choice of operator. The weighted averaging approach is also computationally cheaper and thus more suitable for future application of the data assimilation system to larger problems. The sensing depth of the COSMOS instrument varies with soil moisture but does not extend to the deepest JULES layer (Evans et al., 2016; Antoniou et al., 2019), which is why this layer is excluded from the weighted averaging. However, assimilation updates still reach deeper layers through strong vertical correlations in the background error covariance.

**Reference:**

Evans, J. G., Ward, H. C., Blake, J. R., Hewitt, E. J., Morrison, R., Fry, M., Ball, L. A., Doughty, L. C., Libre, J. W., Hitt, O. E., Rylett, D., Ellis, R. J., Warwick, A. C., Brooks, M., Parkes, M. A., Wright, G. M. H., Singer, A. C., Boorman, D. B., and Jenkins, A.: Soil water content in southern England derived from a cosmic-ray soil moisture observing system – COSMOS-UK, Hydrol. Proc., 30, 4987–4999, https://doi.org/10.1002/hyp.10929, 2016.

Antoniou, V., Askquith-Ellis, A., Bagnoli, S., Ball, L., Bennett, E., Blake, J., Boorman, D., Brooks, M., Clarke, M., Cooper, H., Cowan, N., Cumming, A., Doughty, L., Evans, J., Farrand, P., Fry, M., Hewitt, N., Hitt, O., Jenkins, A., Kral, F., Libre, J., Lord, W., Roberts, C., Morrison, R., Parkes, M., Nash, G., Newcomb, J., Rylett, D., Scarlett, P., Singer, A., Stanley, S., Swain, O., Thornton, J., Trill, E., Vincent, H., Ward, H., Warwick, A., Winterbourn, B., and Wright, G.: COSMOS-UK user guide: users' guide to sites, instruments and available data (version 2.10), Tech. Rep.,Wallingford, http://nora.nerc.ac.uk/id/eprint/524801/, 2019.

5) Furthermore, the error estimate for the observations does not account for the relation between hourly values and a daily value for this cumulative measurement nor the Poisson distribution of its uncertainty instead of a Gaussian.

The analysis of the data showed that the standard deviation of the hourly values around the daily mean is approximately 20% (as mentioned in lines 258–262). However, we have inflated the observation error to 50% of the daily mean observation to account for multiple sources of uncertainty, including the conversion of neutron counts to soil moisture, the averaging of hourly measurements to a daily scale, and temporal (intra-site) observation error correlations arising from persistent calibration or environmental effects at a given site. We also recognize the presence of representation errors (Janjić et al., 2018), which may result from differences in scale between the model and the observations, or simplifications in the observation operator.

To compensate for these unaccounted error sources, we adopt an inflation approach for the observation error variance. Similar techniques have been used in other data assimilation contexts, such as in the assimilation of satellite radiances for numerical weather prediction, where inflation is applied to account for unmodelled observation error correlations (Liu et al., 2003; Stewart et al., 2008).

Regarding the use of Gaussian error probability density functions (pdfs), these are commonly used in data assimilation even though real-world pdfs may be non-Gaussian (Fowler and van Leeuwen, 2013). The Gaussian assumption leads to a quadratic loss function and a mathematically tractable minimization problem. Alternative pdfs may not preserve this property, particularly when combining different distributions with Bayes' rule. Furthermore, in our experiments, the distribution of representation errors is unknown, and the Gaussian pdf is a reasonable choice as it is the maximum entropy distribution for a given mean and covariance.

**Reference:**
Liu, Z. Q., & Rabier, F. (2003). The potential of high-density observations for numerical weather prediction: A study with simulated observations. *Quarterly Journal of the Royal Meteorological Society: A journal of the atmospheric sciences, applied meteorology and physical oceanography*, *129*(594), 3013-3035.

Stewart, L. M., Dance, S. L., & Nichols, N. K. (2008). Correlated observation errors in data assimilation. *International journal for numerical methods in fluids*, *56*(8), 1521-1527. https://doi.org/10.1002/fld.1636

Janjić, T., Bormann, N., Bocquet, M., Carton, J. A., Cohn, S. E., Dance, S. L., Losa, S. N., Nichols, N. K., Potthast, R., Waller, J. A., & Weston, P. (2018). On the representation error in data assimilation. *Quarterly Journal of the Royal Meteorological Society*, 144(713), 1257–1278. https://doi.org/10.1002/qj.3130

6) In respect to the definition of the Cosby's pedotransfer functions, there are also shortcomings. The manuscript refers to Cosby et al. (1984), but not everything presented is given there and seems neither

developed within the manuscript's study. Cosby et al. (1984) has reported linear relations between grain fractions and four hydraulic variables, but not the full mathematical equations as presented in 2.3. Therefore, a part of the earlier development seems to be missing. Marthews et al. (2014) could be cited *directly* in this respect, as one component. But further considerations would be helpful.

We agree with the reviewer that the full set of equations presented in Section 2.3 are not entirely as given in Cosby et al. (1984). We will revise the manuscript to directly cite Marthews et al (2014)

7) And some discussion, why this set of pedotransfer functions? Only because they have been used in the similar preceding study (Cooper et al., 2021a)? And why not start with the parameters adjusted there already? How does it compare to other pedotransfer functions as used in other studies, etc.

We chose the Cosby et al. (1984) PTFs because they are widely used, simple to apply, and commonly used with the JULES land surface model. Their formulation provides continuous functions that are well-suited for representing spatial heterogeneity in soil properties across large scale. One of the key advantages of Cosby's approach is that it relies only on soil texture information, making it a practical and efficient method, especially when direct measurements of hydraulic conductivity are unavailable. Furthermore. Lee (2005) compared different PTFs for estimating soil hydraulic conductivity and found that Cosby's PTF provided the best prediction of saturated hydraulic conductivity.

Another important reason for choosing Cosby's PTFs is their planned operational use by the UK Met Office in their land surface modelling framework. Using the same PTFs ensures our findings are aligned with practical applications and relevant for future operational use.

The reason for not choosing the parameters already estimated by Cooper et al (2021a) as a starting point is explained in our response to comment 2.

8) The title is full of abbreviations and unclear

We acknowledge the reviewer's concern about the use of abbreviations in the title. To improve clarity, we will revise the title to: *"Improving Land Surface Model Soil Moisture Estimates through Hybrid Data Assimilation of In-Situ Soil Moisture Observations."*

9) The introduction to monitoring of soil moisture starts with a general list of remote sensing sensors (and references) and rather outdated observation networks reported in 2006 and 2007. This part could be more to the point and up to date.

We will revise the introduction to make sure it is succinct and strictly relevant to this paper. We will incorporate more recent in-situ soil moisture networks alongside the existing ones to ensure the discussion remains up to date. These additions include the Murrumbidgee Soil Moisture Monitoring Network (MSMMN) (Smith et al., 2012), FR-AQUI (Aquitaine soil moisture network, France) (Jean-Pierre W, et al., 2018), the Center for Western Weather and Water Extremes (CW3E) network (Sumargo et al., 2021), and the GROW Observatory (Xaver et al., 2020).

**References:**

Smith, Adam B., Jeffrey P. Walker, Andrew W. Western, R. I. Young, K. M. Ellett, R. C. Pipunic, R. B. Grayson, Lionel Siriwardena, Francis HS Chiew, and Harald Richter.: The Murrumbidgee soil moisture monitoring network data set, Water Resources Research, 48,7, https://doi.org/10.1029/2012WR011976, 2012.

Wigneron, Jean-Pierre, Sylvia Dayan, Alain Kruszewski, Christelle Aluome, Marie Guillot-Ehret Amen AI-Yaari, Lei Fan, Serhat Guven et al.: The aqui network: soil moisture sites in the "Les landes" forest and graves vineyards (Bordeaux aquitaine region, France), In IGARSS 2018-2018 IEEE International Geoscience and Remote Sensing Symposium, pp. 3739-3742. IEEE, https://doi.org/10.1109/IGARSS.2018.8517392, 2018.

Xaver, Angelika, Luca Zappa, Gerhard Rab, Isabella Pfeil, Mariette Vreugdenhil, Drew Hemment, and Wouter Arnoud Dorigo.: Evaluating the suitability of the consumer low-cost Parrot Flower Power soil moisture sensor for scientific environmental applications, Geoscientific Instrumentation, Methods and Data Systems, 9, 117-139, no. 1, https://doi.org/10.5194/gi-9-117-2020, 2020.

Sumargo, Edwin, Hilary McMillan, Rachel Weihs, Carolyn J. Ellis, Anna M. Wilson, and F. Martin Ralph.: A soil moisture monitoring network to assess controls on runoff generation during atmospheric river events, Hydrological Processes 35, no. 1, https://doi.org/10.1002/hyp.13998, 2021.

10) In line 56 a reference is needed, as such and also for the claim to be more accurate.

We agree that the original statement was a bit unclear and we will revise the sentence, and include references, as
*"Hybrid data assimilation methods, such as Ensemble-Variational approach, combine features of both variational and ensemble techniques by using ensemble-based background error covariances within a variational framework. These methods avoid the need to explicitly compute the full model adjoint while still approximating the model trajectory over an assimilation window, offering a flexible alternative to traditional approaches (Lorenc, 2015; Poterjoy and Zhang, 2015)."*

**References:**

Lorenc, A. C., Bowler, N. E., Clayton, A. M., Pring, S. R., & Fairbairn, D. (2015). Comparison of hybrid-4DEnVar and hybrid-4DVar data assimilation methods for global NWP. *Monthly Weather Review*, *143*(1), 212-229.

Poterjoy, J. and Zhang, F.: Systematic Comparison of Four-Dimensional Data Assimilation Methods With and Without the Tangent Linear Model Using Hybrid Background Error Covariance: E4DVar versus 4DEnVar, Monthly Weather Review, 143, 1601 – 1621, https://doi.org/10.1175/MWR-D-14-00224.1, 2015

11) Line 175: It is a bit uncalled-for to first give p a time index and then declare that it is constant in time.

We appreciate the reviewer's comment and agree that the current wording may be confusing. This will be rectified in the revised manuscript.

- In a data assimilation cycling system, where there are sequential assimilation time-windows and subsequent forecast steps, it is common to express both model states and parameters as functions of time.  In a cyclic data assimilation system, parameters would be updated  after each assimilation window and before the next forecast step, and in this sense they would evolve with time.  Assigning a time index to the parameter vector ensures consistency with the time-dependent structure of the model state equations Thus, our notation has been designed for flexibility for future applications, with cycling in mind.

- In this study, however, we only use one assimilation window. The model parameters are assumed to be time-invariant over the data assimilation time-window, as the properties they represent (e.g., soil hydraulic parameters) are not expected to vary significantly over the one-year time-window considered.

12) Line 245 to 253   Replace $N_e$ by the value chosen here (50), as mentioned anyway several times and as the other particular parameters are also specified as values.

We will replace $N_e$ by 50 at  the three places suggested.

13) The references contain a large number of malformed doi links.

Thank you for pointing this out. We  will correct the DOI formatting.

---

## Author Comment (AC2)

**Improving JULES Soil Moisture Estimates through 4D-En-Var Hybrid Assimilation of COSMOS-UK Soil Moisture Observations**

Response to Anonymous Referee #2 – Visweshwaran et al., (2025) doi:10.5194/egusphere-2024-3980

The authors thank the reviewer for their insightful comments. Below we list each comment and have numbered them for ease of reference. Our responses are provided below in purple. References are as cited in the manuscript or provided in each response.

General points:

1. Since the initial states and parameters are both time-invariant, it would be simplest to consider both as "parameters". This would avoid the confusion about updating states (commonly called "nudging") which is usually associated with the term "data assimilation" but is not happening here. The application here is essentially parameter calibration (which is fine, only the description is confusing). For example "During state assimilation, the initial conditions are updated for each site" (line 286) invites confusion. States are not being assimilated or updated in the usual sense used in the context of data assimilation applied to NWP.

We agree that the terminology could be misinterpreted and will revise the manuscript to clarify that what we refer to as "state assimilation" involves estimation of the initial soil moisture conditions at the start of a single assimilation window. While this constitutes state estimation within a data assimilation framework, we acknowledge that it differs from the data assimilation cycling systems typically used in numerical weather prediction and reanalysis. This distinction will be made clear throughout the revised manuscript.

More broadly, we would like to emphasise that this work is part of a larger , stepwise development of a data assimilation framework to generate a kilometre-scale reanalysis for the UK using the JULES community model. While the present study uses a single-column (1D) configuration to test and validate the methodology at multiple sites using field-scale soil moisture observations, a cycling data assimilation system is currently under development. Hence the terminology we use in this paper has been carefully chosen to also be relevant to the future system.

2. In what sense is this "4D"? There is only one time window, so there is nothing dynamic going on, carrying information across multiple time windows. The observational data form a single time series, but time is not treated as a special dimension with autocorrelation - they appear to considered as independent data. There is no horizontal spatial dimension to the data - the 16 sites are considered separately. Only a single vertical dimension is represented, so is this not "1DVar"?

Strictly speaking, our problem is two-dimensional—one dimension in the vertical and one in time: the ensemble-variational approach in this study is applied using a single-column (1D) JULES model. The cost function includes a sequence of observations distributed over time, and the control vector (initial conditions and parameters) is propagated forward in time within the assimilation window using JULES. Although we do not assimilate observations across multiple cycles, the use of an ensemble of model forecasts links the control vector to a time series of observations, allowing temporal dynamics to inform the assimilation. This is conceptually consistent with the 4D-Var framework,

where the model evolution over time constrains the optimisation. This will be clarified in the Section 2.4. Further, to avoid confusion, we will revise the manuscript to refer to the method more accurately as an "ensemble-variational" approach, rather than using the term "4D-En-Var."

3. Related to the above, variational DA is usually applied to initial value problems, where the model is highly sensitive to its starting state, as in numerical weather prediction (NWP). This is not the case here: the model is clearly not sensitive to its initial values (Fig. 3): the lines converge after 2 months and are indistinguishable for the 22 months thereafter. The difference between SC2 & 3 is almost literally zero in terms of absolute accuracy of prediction (Table 1). (I suspect KGE is inflating the differences by expressing relative differences.)

We agree with the reviewer that the model shows time-limited sensitivity to initial conditions, as illustrated in Figure 3. We have already acknowledged in the manuscript (Lines 291–295) that the effect of updating the initial conditions fades beyond the first three to four months. This information is nevertheless useful in designing a cycling data assimilation system for reanalysis, where identifying the effective-impact-time of observations helps inform the appropriate length of each assimilation cycle. This point will be brought out more clearly in the revised manuscript.

Regarding the comparison between Scenarios 2 and 3 in Table 1, we acknowledge that the overall RMSE differences are small. However, this result is still meaningful. In many sites—such as EASTB, GISBN, MOORH, SOURH, and PORTN—Scenario 3 shows a noticeable improvement over Scenario 2, suggesting that jointly updating the state and parameters provides added benefit at these locations. Additionally, as shown in Figures 6 and 9, the magnitude of hydraulic parameter adjustments is smaller in Scenario 3 compared to Scenario 2. This indicates that incorporating initial conditions helps moderate the parameter updates, resulting in a more constrained and possibly more physically plausible calibration. This behaviour supports the value of joint state-parameter assimilation in improving soil moisture estimates. We will revise the manuscript to reflect these points more clearly in the conclusion section.

4. Given that the application here is essentially parameter calibration, this begs the question of whether "4D-En-Var" is the best method, perhaps over-kill or maybe inappropriate. Would standard optimisation or ML estimates be any better or worse, or would statistical time series analysis be more appropriate? In the light of this, I think it needs some justification as to why this method was chosen, and perhaps some comparison with more obvious choices.

Thank you for the comment. The ensemble-variational framework offers several practical advantages that make it particularly suitable for land surface model applications such as ours.

As shown in recent studies (e.g., Douglas et al., 2025; Beylat et al., 2025), 4D-En-Var provides an efficient and scalable approach for parameter estimation by avoiding the need to develop and maintain adjoint or tangent linear models—something that can be a major obstacle to gradient-descent based optimisation schemes for dynamic models like JULES.

Gradient-free statistical approaches like MCMC, are often computationally prohibitive for high-dimensional systems, as they require many model realisations to effectively sample the probability density. In contrast, the ensemble-variational approach only requires a moderate ensemble size, and enables reusability of model trajectories across experiments.

A machine learning (ML) approach would typically minimize a RMSE loss function which has a similar form to the 4D-En-Var cost function, but without explicit consideration of uncertainties or prior dynamical information. Hence, in ML unlike ensemble-variational data assimilation, very large training datasets are typically needed to avoid overfitting and ensure generalisability.

These characteristics make the ensemble-variational approach a robust and flexible option for building general-purpose, large land data assimilation systems, including the one under development in this study (see comment 1). If we are invited to revise the paper, we will clarify the motivation for our choice of method in the Introduction section.

Refences:

Beylat, S., Raoult, N., Bacour, C., Douglas, N., Quaife, T., Bastrikov, V., ... & Peylin, P. (2025). Towards the Assimilation of Atmospheric CO 2 Concentration Data in a Land Surface Model using Adjoint-free Variational Methods. *EGUsphere*, *2025*, 1-33.

Douglas, N., Quaife, T., & Bannister, R. (2025). Exploring a hybrid ensemble–variational data assimilation technique (4DEnVar) with a simple ecosystem carbon model. *Environmental Modelling & Software*, 106361.

5. The model is referred to as "the JULES community land surface model" but the focus here is purely a calculation of hydraulic conductivity, which is implicitly used to predict soil moisture via Darcy's law (though this is not actually presented). It is not clear if the other components of the JULES model are relevant. It provides a prediction of evapotranspiration, though perhaps there are local site estimates which might be more accurate, using lysimeters or observed LAI & Penman-Monteith etc. To what extent might the model-observation discrepancy be due to bias in the modelled evapotranspiration?

In this study, the focus is specifically on the soil hydraulic characteristics module of the JULES model, as it governs the vertical redistribution of soil moisture through the profile. Other components of the JULES land surface model—such as the surface hydrology, canopy interception, and runoff schemes—are retained as part of the full model structure but are not directly examined here. They operate as upstream processes that determine the water input available to the soil column and therefore serve as inputs to the hydraulic module evaluated in this work. In JULES, surface processes such as canopy interception, surface runoff, and throughfall are governed by the surface water balance equations described in Best et al. (2011). Once water reaches the soil surface, its vertical movement through the layers is calculated using the finite-difference form of Richards' equation (as described in Equation 1 of the manuscript), based on the soil hydraulic parameters derived from pedotransfer functions. These details will be briefly included in Section 2.2.

Regarding evapotranspiration (ET), we agree that uncertainty in the ET component—calculated in JULES using the Penman-Monteith formulation—can influence modelled soil moisture independently of the soil hydraulic properties. However, we do not currently have independent, co-located ET observations (e.g., from lysimeters or flux towers) for assimilation. We acknowledge this as a limitation, and will include a note in the discussion to suggest that future work could benefit from incorporating ET data to further reduce model–observation discrepancies.

6. Using the HWSD instead of the locally measured soil texture seems an odd choice. A simpler interpretation of the results might be that the Cosby model parameters are fine, but the soil texture estimates from HWSD are wrong (as they will be for any point location), and the calibration results are simply the model parameters counter-balancing incorrect inputs. A comparison using the locally measured soil texture would discriminate between these options.

We agree that HWSD soil texture data may not precisely match site-specific measurements. However, we chose to use the HWSD dataset to ensure consistency across sites and to reflect a realistic scenario for applying this framework in regions where local texture measurements are not available. As mentioned in our response to Comment 1, this work is part of a broader effort to implement the methodology across the whole UK using the JULES model, where access to site-specific soil data may be limited. While soil texture measurements are not available for most of the COSMOS-UK

sites, even if such data were available, they would typically represent point-scale conditions. In contrast, both the JULES simulation and the COSMOS-UK observations represent field-scale processes, and such scale mismatches could introduce additional inconsistencies if point-scale texture values were used directly.

Several global soil texture datasets exist (e.g., SoilGrids; Hengl et al., 2017), and while discrepancies with local measurements have been documented (e.g., Zhao et al., 2018), HWSD has been used successfully in previous JULES studies (e.g., Martínez-de la Torre et al., 2019; Ritchie et al., 2019). In particular, Cooper et al. (2021a) demonstrated that HWSD-derived soil texture data can be effectively integrated into the JULES data assimilation framework, leading to improved soil moisture estimates. Building on this prior success and given its relevance to large-scale and operational applications, HWSD was chosen for this study.

Nevertheless, to account for potential inaccuracies in these inputs, we inflated the observation error statistics within the data assimilation framework. This inflation helps to absorb uncertainties arising not only from the COSMOS-UK observations but also from the HWSD-derived soil texture values.

References:

Hengl, T., Mendes de Jesus, J., Heuvelink, G. B. M., Ruiperez Gonzalez, M., Kilibarda, M., Blagotíc, A., Shangguan, W., Wright, M. N.,Geng, X., Bauer-Marschallinger, B., Guevara, M. A., Vargas, R., MacMillan, R. A., Batjes, N. H., Leenaars, J. G. B., Ribeiro, E., Wheeler,I., Mantel, S., and Kempen, B.: SoilGrids250m: Global gridded soil information based on machine learning, PLOS ONE, 12, 1–40, https://doi.org/10.1371/journal.pone.0169748, 2017.

Martínez-de la Torre, A., Blyth, E. M., and Weedon, G. P.: Using observed river flow data to improve the hydrological functioning of the JULES land surface model (vn4.3) used for regional coupled modelling in Great Britain (UKC2), Geoscientific Model Development, 12,765–784, https://doi.org/10.5194/gmd-12-765-2019

Ritchie, P. D. L., Harper, A. B., Smith, G. S., Kahana, R., Kendon, E. J., Lewis, H., Fezzi, C., Halleck-Vega, S., Boulton, C. A., Bateman,I. J., and Lenton, T. M.: Large changes in Great Britain's vegetation and agricultural land-use predicted under unmitigated climate change, ENVIRONMENTAL RESEARCH LETTERS, 14, https://doi.org/10.1088/1748-9326/ab492b, 2019.

Zhao, H., Zeng, Y., Lv, S. & Su, Z. 2018, Analysis of soil hydraulic and thermal properties for land surface modeling over the Tibetan Plateau, Earth system science data. 10, 2, p. 1031

7. Essentially the algorithm calibrates 12 parameters, which relate soil texture to hydraulic properties, and so we are indirectly calibrating V_sat, b, sathh and satcon. How do the default & calibrated estimates of V_sat (i.e. porosity, easily calculated from bulk density) relate to the locally measured values at the COSMOS sites? This would be a more informative analysis, telling us whether the DA has produced a more accurate estimate of a mechanistically important and measurable parameter. As it stands, the paper is basically reporting that optimising the parameters to minimise RMSE improves RMSE (this is essentially what the cost function does, albeit weighted and penalised towards the defaults). That in itself is rather circular, so requires some analysis of *how* the parameters are changed, whether this is meaningful or arbitrary.

Thank you for the comment. We would like to highlight that in the submitted manuscript we discuss the physical interpretation of how the JULES hydraulic parameters have changed, and the effect this has on soil moisture forecasts, in the results sections for both Scenario 2 and Scenario 3 (Sections 3.2 and 3.3, respectively). These sections explain how the posterior parameters shift relative to the priors and describe their influence on the model behaviour. This addresses the "how" aspect raised by the reviewer and shows that the changes are not arbitrary but relate to model performance and physical plausibility. However, we acknowledge that the existing discussion in Section 3 may be a little buried and we will rephrase this section to give it more emphasis. We also note that the performance

improvements persist beyond the 2017 assimilation period into 2018, when no observations were assimilated. This suggests that the updated parameters enhance model generalisation rather than simply overfitting to the assimilation window. We will revise Section 3 to give these points greater prominence in the revised manuscript.

8. Are data for b, sathh and satcon available from the COSMOS sites? Soil water retention curves are fairly standard measurements for soil physics/hydrology. As above, the comparison between the measured values and those produced after model calibration would be of interest. Indeed, more interest, since this might reveal something of the underlying mechanisms.

Unfortunately, such site-specific soil hydraulic measurements (e.g., $\theta_{sat}$, $b$, $\psi_{sat}$, $K_{sat}$) are not available for all the sites and time periods used in this study. Even if these measurements were available, they would typically be collected at the point scale, whereas our study focuses on estimating field-scale parameters. Due to this inherent scale mismatch, a direct and fair comparison would not be feasible.

9. To the non-JULES user, the distinction between the Cosby model and the JULES model is very arbitrary. For practical purposes, you are calibrating 12 model parameters. Referring to these as a separate "pedotransfer function" and PTF constants does not help explain conceptually what is going on here.

We distinguish here between the pedotransfer function (PTF) model and the JULES land surface model. JULES includes a soil hydraulic module that requires key parameters such as porosity ($\theta_{sat}$), saturated matric potential ($\psi_{sat}$), and saturated hydraulic conductivity ($K_{sat}$). These parameters cannot always be directly measured at the required spatial scale. To overcome this, PTFs—empirical models such as the one proposed by Cosby et al. (1984)—are commonly used to derive these hydraulic parameters from more readily available soil texture data (e.g., sand, silt, clay fractions). The Cosby PTF is not part of JULES itself, but is applied as a preprocessing step to estimate inputs for JULES from soil texture maps. Other options for the PTFs are available (Wösten et al., 1999; Schaap et al., 2004; Tóth et al., 2015) but we have chosen the Cosby PTF model here to facilitate direct comparison with previous work Cooper et al., (2021a).

In this study, we optimise the constants of the Cosby PTF ($K1$–$K12$), not the hydraulic parameters directly. This allows us to generate soil physics parameters that are internally consistent and aligned with the structure of the Cosby model. Directly updating individual parameters like $\theta_{sat}$ or $K_{sat}$ could break the physical relationships defined by the PTF system, potentially introducing inconsistencies. We will revise Section 2.3 to clarify this distinction and help readers unfamiliar with JULES understand how the PTF model interfaces with the land surface model.

References:

Wösten, J., Lilly, A., Nemes, A., and Le Bas, C.: Development and use of a database of hydraulic properties of European soils, Geoderma, 90, 169–185, https://doi.org/10.1016/S0016-7061(98)001323, 1999.

Schaap, M. G., Nemes, A., and van Genuchten, M. T.: Comparison of Models for Indirect Estimation of Water Retention and Available Water in Surface Soils, Vadose Zone J., 3, 1455–1463, https://doi.org/10.2136/vzj2004.1455, 2004.

Tóth, B., Weynants, M., Nemes, A., Makó, A., Bilas, G., and Tóth, G.: New generation of hydraulic pedotransfer functions for Europe, Eur. J. Soil Sci., 66, 226–238, https://doi.org/10.1111/ejss.12192,2015.

Specific points:

l108. 50 sites? Why use only 16? Computational reasons? How were the 16 sites chosen?
Sites were primarily chosen based on the availability of continuous driving data over the assimilation and forecast period while also seeking to ensure a representative range of soil types, including typical mineral soils, high organic content soils, and calcareous mineral soils. This selection also allows for a fair and consistent comparison with the work of Cooper et al. (2021a), which used the same set of sites. Using the same locations ensures that differences in results are due to methodological developments rather than site-specific variability.

l123. Eqn 1: this is just mass balance, not the Richards' equation. Perhaps leave out the R_bl runoff term since you assume it to be zero, so as not to mislead. Where does precipitation fit in? You show the outputs but not the input.
We agree that Equation (1) represents a layer-wise water balance derived from the finite difference form of the Richards equation, and not the Richards equation itself. The term $R_{bl}$ is indeed set to zero in our study, as mentioned in the text (L127), and we will remove it from the equation to avoid confusion. As pointed out in our response to comment 5, we will clarify that precipitation is not directly used in this equation; rather, the input to the soil column is the infiltrated water that remains after canopy interception and surface runoff, as governed by the surface scheme in JULES (Best et al., 2011, Equations 46–49). We will revise the text accordingly to better reflect this.

l124, 128, Eqns 1-3. "soil moisture content" can be defined in many ways. I presume we are talking here about volumetric content in m^3 / m^3, since that is what CNRS is sensitive to. Please define as such, with units.

The unit of soil moisture content (m³/m³) will be explicitly defined in the revised manuscript.

l131+ throughout. Assuming the above is correct "soil moisture content" is denoted with three different but duplicate symbols: theta, V and SM. Why? This is just horrendously confusing. Just use one - theta is fairly standard. In Eqns 2. "soil moisture content" is defined as theta but its value in saturated conditions is defined with a different symbol, V. This obfuscates the point that the LHS is a fraction-of-the-whole term, i.e. soil moisture content as a fraction of its maximum possible value.
l131+ throughout, Eqns 1-3. Similarly, matric suction is defined as psi, but its value in saturated conditions is defined with as "sathh", producing the same obfuscation as above. (And multi-letter symbols are generally frowned up.) Units?
l131+ throughout, Eqns 1-3. Similarly, hydraulic conductivity is defined as K, but its value in saturated conditions is defined with as "satcon", producing the same obfuscation as above.
Thanks for pointing this out. We will revise the manuscript to use consistent notation throughout: soil moisture content will be denoted as $\theta$, matric suction as $\psi$, and hydraulic conductivity as $K$. The symbols for their respective saturated values will be changed to $\theta_{sat}$, $\psi_{sat}$, and $K_{sat}$ and the relevant units given. This will be applied across all equations and text to avoid confusion.

l146, Eqn 4. K has already been defined as hydraulic conductivity, and now it is given a different meaning! Use different symbols for different quantities, (but the same symbol when referring to the same quantity).
To avoid confusion with the previously defined symbol for hydraulic conductivity, the PTF constants will be denoted using "$C$" throughout the manuscript. This change will be applied consistently across all relevant equations, text, and figures.

l150. V_sat is defined as the maximum "water-holding capacity of the soil". Given the different wording, It's not clear if this actually refers to volumetric soil moisture content or not.

l158. Same points apply to V_crit and V_wilt.

We agree that the wording could be clearer. In the revised manuscript, we will explicitly state that $V_{sat}$ refers to the volumetric water content at saturation, which is the maximum amount of water the soil can hold per unit volume when fully saturated. Similarly, the definitions of $V_{crit}$ and $V_{wilt}$ be clarified to indicate that they represent the volumetric soil moisture content corresponding to matric suctions of –33 kPa (field capacity) and –1500 kPa (wilting point), respectively.

Eqn 8. Is the denominator 3.364 related to -33 kPa? Make the link clear if so; explain if not.

Eqn 9. Is the denominator 152.9 related to -1500 kPa? Make the link clear if so; explain if not.

Yes, Equations (8 and 9) are derived from Equation 2 at fixed values of matric suction corresponding to the critical and wilting points. This will be added to the revised manuscript.

l158. Are V_crit and V_wilt relevant? They are only mentioned once hereafter, and no field measurements are presented for comparison.

Yes, $V_{crit}$ and $V_{wilt}$ are relevant in this study as they help assess changes in plant-available water following data assimilation and are part of the system of pedotransfer functions that we seek to optimise in this work. The $V_{crit}$ and $V_{wilt}$ thresholds reflect key points on the soil moisture retention curve and provide insight into how soil water availability shifts. While they are briefly discussed in the results section (after Figures 6 and 9, around Line 311), we agree that their significance is not sufficiently explained in the current version of the manuscript. We will revise Section 2.3 to provide a clearer explanation of why they are useful in the context of this study. As noted in our response to Comment 8, we do not have local in-situ measurements of these values for direct validation.

l164. Is the soil heat storage relevant here? It is not mentioned again I think.

The variables for soil heat capacity ($h_{cap}$) and soil thermal conductivity ($h_{con}$) are relevant because they are derived from the same Cosby PTFs and influence the thermal regime of the soil, which in turn can affect soil moisture dynamics indirectly through evapotranspiration. Although the current study focuses primarily on soil moisture, we will revise Section 2.3 to briefly explain the relevance of them in this study.

Section 2.3/4. We never return to Eqn 1 to explain how the derived parameters are used to calculate the water flux between vertical layers. How do the equations and parameters presented actually result in a prediction of soil moisture in different layers. Where does precipitation fit in?

We acknowledge that the connection between the derived hydraulic parameters and the calculation of soil moisture within JULES could be explained more clearly. In the revised manuscript, we will expand Section 2.3 to briefly clarify this process. The water flux between layers in JULES is computed using Darcy's law, given by:

$$W' = K_h \left\{ \frac{\partial \psi}{\partial z} + 1 \right\}$$

where $W'$ is the vertical water flux, $K_h$ is the unsaturated hydraulic conductivity, and $\psi$ is the soil matric potential. The gradient $\frac{\partial \psi}{\partial z}$ is computed between adjacent layers.

These layer-to-layer fluxes form the core of the finite-difference implementation of the Richards equation used in JULES to update soil moisture over time. The derived hydraulic parameters—calculated via the Brooks and Corey relationships using the PTF constants—control how moisture content and hydraulic conductivity vary with soil water potential, and thus determine the magnitude of these vertical fluxes.

Precipitation enters the system through the surface hydrology module, which determines the amount of water infiltrating into the soil. This infiltrated water serves as the upper boundary condition for the Richards equation. The lower boundary condition is defined as free drainage. These aspects are governed by the surface water balance equations in Best et al. (2011), adding to our previous response to Comment 5. While we have not focused on analysing the layer-to-layer fluxes in this study, we note that our primary interest is in the resultant soil moisture profiles for comparison with the COSMOS-UK observations. Section 2.3 of the manuscript will be revised accordingly to clarify this flow of information from precipitation input to vertical moisture redistribution via the derived parameters and governing equations.

l178+. It is clearly stated that the parameters do not vary in time. Why then are they shown throughout with the subscript t, which implies precisely that they *do* vary in time? Very confusing.

We appreciate the reviewer's comment and agree that the current wording may be confusing. This will be rectified in the revised manuscript.

In a data assimilation cycling system, where there are sequential assimilation time-windows and subsequent forecast steps, it is common to express both model states and parameters as functions of time. In a cycling data assimilation system, parameters would be updated after each assimilation window and before the next forecast step, and in this sense they would evolve with time. Assigning a time index to the parameter vector ensures consistency with the time-dependent structure of the model state equations. Thus, our notation has been designed for flexibility for future applications, with cycling in mind.

In this study, however, we only use one assimilation window. The model parameters are assumed to be time-invariant over the data assimilation time-window, as the properties they represent (e.g., soil hydraulic parameters) are not expected to vary significantly over the one-year time-window considered. This will be highlighted in Section 2.4.

l172+, Section 2.4. Can you describe in words for the non-expert what the algorithm is doing. The description here is couched in the terminology of 4D-Var, which is not self-explanatory for the general readership of HESS. e.g. I think Eqn 12 just says that the predicted states at time t are modelled as a function of their states at t-1 and some parameters p, but that is not obvious to the casual reader because of the language used.

While we have provided some explanation of the En-Var components for this study, we agree that it can be explained more clearly for readers outside the data assimilation community. Section 2.4 will be revised accordingly to provide a more intuitive, step-by-step description of what the algorithm is doing. Additionally, two schematics will be added to illustrate the terms of the EnVar cost function and the data assimilation process.

l88+. "encapsulates" and "such as" implies there are other variables as well. I don't think that is so, but needs clarifying. The control vector includes parameters, not just "variables" as stated here.
We will revise the sentence in Line 188 to clarify that the control vector includes either the initial soil

moisture states, the 12 Cosby PTF constants, or both—depending on the assimilation scenario. The revised manuscript will read as "This optimization is achieved by estimating the control vector, $x_t$, which, in our implementation, includes either the initial soil moisture states, the 12 Cosby PTF constants, or both—depending on the assimilation scenario"

l191. You don't "aim to estimate the initial soil moisture states" - these are not the quantities of interest. These are the initial states that are allowed to vary / be updated.

The sentence will be revised

l192. By definition, the initial values do not vary in time. Why then are they shown throughout with the subscript t, which implies precisely that they do vary in time?

While we only estimate the state at the start of the assimilation window (the initial value when $t = 0$), the ensemble is evolved in time through the assimilation window, ensuring that the assimilation fits the observations along the model trajectory. Thus, using a time index for the state vector is natural.

l93. Hazarding a guess at what is actually going on, contrary to what is written, I think x_a is the "control" vectors of parameters which are varied in the algorithm, whilst x_t seems to be the states that are compared with observations, and are thereby different things given the same symbol "x". I think x_t = [z_t]^T in all scenarios, i.e. it is always the current soil moisture states that are compared with observations. It is the make-up of x_a and x_b that differ among scenarios, whether they consist of initial soil moisture states, soil parameters, or both.

As explained at the beginning of Section 2.4, $z$ denotes the state vector and the $p$ parameter vector. The control vector, $x$, represents the variables being optimized, and its composition differs across scenarios. For example, in Scenario 1 (state-only assimilation), the control vector consists solely of the initial soil moisture states, i.e. $x = [z]^T$. In Scenario 3 (joint state-parameter assimilation), the control vector includes both initial states and soil parameters, i.e., $x = [p\ z]^T$. We will rephrase this section for greater clarity.

l200. Don't add synonyms for no reason - use either "background", "before" or "prior"; and "analysis", "after" or "posterior". I'd avoid the last to avoid confusion with Bayesian estimates (this seems to be a penalised maximum likelihood estimate).

We will use consistent terminology throughout the manuscript and retain the terms "prior" and "posterior," as the estimates in this study are derived within a Bayesian framework. These terms will be thoroughly explained when they are introduced.

l181. The "observation operator" h_t() requires explanation. How does it "map the state vector to the observation space"? What does this mean? Subscript t implies that the function h varies in time. Explain how this is so.
l201-217. Explain what the "background error covariance matrix" is in plain English. In this context, it seems to quantify the uncertainty in the default parameters. How then can it be estimated from "a previous forecast" (line 208) since we can never forecast parameters or initial state values? The explanation in terms of the perturbation matrix, x'_b does not make sense to me (perhaps my maths limitations). "Ensemble members" are mentioned without any explanation of what this is at this point.

l203, Eqn 14. Can you give a plain English explanation of the cost function? Looks like a weighted average of (i) deviation of parameters from their default values and (ii) deviation of predictions from

observed values, with the weighting coming from the error matrix which indicates how much uncertainty we should attach to each.

We will revise Section 2.4 to make the various components of the data assimilation method clearer for a general audience. This includes explaining the following:

- **Observation operator $h_t(\cdot)$**: This maps the model state (soil moisture at different layers) into the model-equivalent of the observation space (COSMOS-UK measurements) so that the model output can be compared to observed data.

- **Ensemble members (L232-L235)**: These are a set of sampled versions of the control vector, generated by adding random perturbations to the prior values. This ensemble provides the variability needed to estimate uncertainty and calculate gradients in the cost function. Each ensemble member corresponds to a separate JULES model run, initialised using a different realisation of the PTF constants and/or with a different initial condition. The resulting spread in soil moisture trajectories after the model spin-up reflects the uncertainty associated with the parameters and/or initial conditions.

- **Prior error covariance matrix (B)**: This quantifies the uncertainty in the prior estimates of the control vector (for instance during scenario 3 the control vector includes initial soil moisture condition and PTF constants). In our study, it is estimated using an ensemble of perturbed prior values.

- **Cost function (Equation 14)**: This is a weighted function we minimise during assimilation. It includes two main components: (i) the difference between the control vector and its prior estimate, and (ii) the difference between modelled outputs and observations. Each component is weighted by the inverse of its respective error covariance matrix, which controls the relative influence of observations and prior information in the final solution.

These clarifications will be added in the revised version of Section 2.4. In addition, a schematic of En-Var will be added to visually summarise the process involved (Please refer to the response of l172).

l224. "The posterior estimate" is easily confused with "posterior" in the Bayesian sense, but I don't think that is what is meant here. Possibly this is just the terminology used in 4DVar, but I think this is essentially a penalised maximum likelihood estimate). Better to refer to it as something like "optimised" or clarify how this sits with the Bayesian sense of estimating the posterior distribution if it is actually meant in that sense.

The 4D-Var cost function can be derived from Bayes rule and the resulting analysis is a Maximum a Posteriori (MAP) estimate. We will add comments to the text and cite Asch et al. (2016, Chapter 2) that explains this.

Refences:

Asch, M., Bocquet, M., & Nodet, M. (2016). *Data assimilation: methods, algorithms, and applications*. Society for Industrial and Applied Mathematics. SIAM. Chapter 2.

l233. Are the 12 parameters independent? Probably not. Is this covariance represented in the B matrix?

The 12 parameters are not strictly independent, and their error correlations are captured in the prior error covariance matrix, which is generated from an ensemble of perturbations applied to the prior parameter set. This will be explained in Section 2.4.

l234. This seems to be a non sequitur as a justification of guessing +/- 10%.

The ±10% range for generating these perturbations was selected as a practical estimate to introduce variability around the prior values, based on prior experience with JULES and similar studies (e.g., Cooper et al., 2021a). We will add this information to the manuscript.

l236. HWSD is coarse and rather unreliable for estimates at a specific point. Why not use the locally measured soil texture values, or better, a comparison of sensitivity among different estimates of these inputs?
Please see response to Comment 6.

l245. So are the "initial values" considered to be 1 Jan 2016 or 1 Jan 2017? The latter would make the spin-up irrelevant. I'm confused.
The initial soil moisture states considered for state and state-parameter estimation are from $1^{st}$ Jan 2017 (i.e. after the spin-up period 2016). The purpose of the spin-up period is twofold: first, to derive an ensemble of initial soil moisture values in a reasonably appropriate range. Second, to bring the modelled soil moisture into a quasi-equilibrium state before the simulation period begins.

l260. sigma = 50% in the observations, so these are highly uncertain, and not a strong constraint on the model. This is worth highlighting.

It is clear from the figures 3, 7, and 11 in the manuscript that the posterior is markedly different from the prior. While the observation uncertainty is large relative to the ensemble spread, assimilation of a continuous time series of observations (as done here using En-Var) has a cumulative constraining effect allowing the system to exploit temporal structure in both the model and observations (Lorenc, 2003).

We have inflated the observation error to 50% of the daily mean observation to account for multiple sources of observation uncertainty, including the conversion of neutron counts to soil moisture, the averaging of hourly measurements to a daily scale, and temporal (intra-site) observation error correlations arising from persistent calibration or environmental effects at a given site. We also recognize the contribution of representation errors to observation uncertainty (Janjić et al., 2018), which may result from differences in scale between the model and the observations, or simplifications in the observation operator. To compensate for these unaccounted error sources, we adopt an inflation approach for the observation error variance. Similar techniques have been used in other data assimilation contexts (e.g. Liu et al., 2003; Stewart et al., 2008).

This information will be highlighted in the revised manuscript.

Reference:

Lorenc, A. C. (2003). The potential of the ensemble Kalman filter for NWP – a comparison with 4D-Var. Quarterly Journal of the Royal Meteorological Society, 129(595), 3183–3203. https://doi.org/10.1256/qj.02.132

Stewart, L. M., Dance, S. L., & Nichols, N. K. (2008). Correlated observation errors in data assimilation. International journal for numerical methods in fluids, 56(8), 1521-1527. https://doi.org/10.1002/fld.1636

Janjić, T., Bormann, N., Bocquet, M., Carton, J. A., Cohn, S. E., Dance, S. L., Losa, S. N., Nichols, N. K., Potthast, R., Waller, J. A., & Weston, P. (2018). On the representation error in data assimilation. Quarterly Journal of the Royal Meteorological Society, 144(713), 1257–1278. https://doi.org/10.1002/qj.3130

l269. Which "modelled soil moisture values" are used? There are four layers but only the top two

correspond to the CNRS measurement, depending on soil wetness. Implied that some weighting is done but not explained.

We have adopted a weighted depth approach to align the modelled soil moisture from the four JULES soil layers with the effective sensing depth of the COSMOS-UK observations. The sensing depth of the COSMOS instrument varies with soil moisture but typically does not extend as deep as the bottom layer of JULES (Evans et al., 2016; Antoniou et al., 2019), which is therefore excluded from the weighted averaging. Assimilation updates, however, still influence deeper layers due to strong vertical correlations represented in the background error covariance matrix. The same methodology was used in (Cooper et al., 2021a; Pinnington, E., 2021). These details are already provided briefly in the manuscript at Lines 256-261, but we will rephrase them for greater clarity in the revised manuscript.

**Reference:**

Evans, J. G., Ward, H. C., Blake, J. R., Hewitt, E. J., Morrison, R., Fry, M., Ball, L. A., Doughty, L. C., Libre, J. W., Hitt, O. E., Rylett, D., Ellis, R. J., Warwick, A. C., Brooks, M., Parkes, M. A., Wright, G. M. H., Singer, A. C., Boorman, D. B., and Jenkins, A.: Soil water content in southern England derived from a cosmic-ray soil moisture observing system – COSMOS-UK, Hydrol. Proc., 30, 4987–4999, https://doi.org/10.1002/hyp.10929, 2016.

Antoniou, V., Askquith-Ellis, A., Bagnoli, S., Ball, L., Bennett, E., Blake, J., Boorman, D., Brooks, M., Clarke, M., Cooper, H., Cowan, N., Cumming, A., Doughty, L., Evans, J., Farrand, P., Fry, M., Hewitt, N., Hitt, O., Jenkins, A., Kral, F., Libre, J., Lord, W., Roberts, C., Morrison, R., Parkes, M., Nash, G., Newcomb, J., Rylett, D., Scarlett, P., Singer, A., Stanley, S., Swain, O., Thornton, J., Trill, E., Vincent, H., Ward, H., Warwick, A., Winterbourn, B., and Wright, G.: COSMOS-UK user guide: users' guide to sites, instruments and available data (version 2.10), Tech. Rep.,Wallingford, http://nora.nerc.ac.uk/id/eprint/524801/, 2019.

Pinnington, E., Amezcua, J., Cooper, E., Dadson, S., Ellis, R., Peng, J., Robinson, E., Morrison, R., Osborne, S., and Quaife, T.: Improving soil moisture prediction of a high-resolution land surface model by parameterising pedotransfer functions through assimilation of SMAP satellite data, Hydrology and Earth System Sciences, 25, 1617–1641, https://doi.org/10.5194/hess-25-1617-2021, 2021.

l271, Eqn 21. How does this differ from h_t (x_t ) − y_t in Eqn 14? Or is just a different symbol for soil moisture?

Both expressions involve the same underlying misfit between model and observations, but their purposes differ: $y_t - h_t(x_t)$ is weighted according to the observation uncertainty within the assimilation cost function, while RMSE is used (with no uncertainty weighting) post-assimilation to evaluate model performance during both 2017 and 2018.

l285. effect on "JULES performance". Semantics perhaps, but I think here we are looking at the effect of the performance of the calibration algorithm.

We refer the reviewer to our earlier response regarding the use of the term "state estimation" rather than calibration (Question 1). We will rephrase this in terms of the "skill" of JULES forecasts, following assimilation updates to the initial conditions.

Fig. 3. The model is clearly not sensitive to its initial values: the lines converge after 2 months and are indistinguishable thereafter. This is very clear from Table 1 SC2 vs SC3, and this result needs to be emphasised. How is this reconciled with the distributions in Fig 3 and the RMSE values in Table 1 (Prior vs SC1)? At both sites, layer 3 has the opposite trend to the rest. Is there some kind of numerical artefact going on here? KGE gives the wrong impression, being based on relative differences when it is the absolute difference that matters.

We refer the reviewer to our response to Comment 3, where we have discussed the sensitivity of the model to initial condition updates and the implications for Scenario 3 results (Table 1).

Regarding Prior vs SC1: The reduction in RMSE is primarily due to improved performance during the early stages of the simulation, when the updated initial state has the most influence. Although this benefit diminishes over time, it still contributes to a lower RMSE when averaged across the full period. This understanding is also important for our ongoing work in developing a cycling assimilation system, where the frequency of state updates can be designed based on the duration of their impact.

Regarding the layer 3 distributions in Figure 3, we note that while the posterior means for layers 1, 2, and 4 shifted to slightly higher values, layer 3 shows only minimal change. However, across all layers, the posterior distributions become narrower compared to the prior, indicating a meaningful reduction in uncertainty rather than any numerical artefact. The differences in the prior and posterior values are consistent across the 16 sites, as shown in the appendix.

Finally, we agree that RMSE provides an important measure of absolute error. While KGE is included for completeness, we will ensure that key conclusions are also supported by RMSE-based analysis.